# A two-step mechanism for sugar translocation

Do-Hwan Ahn[1], Claudia Alleva [1,2], Tom Reichenbach[1], Ashutosh Gulati [1], Alessandro Ruda[3], Marta Bonaccorsi[1,2], Jakob M. Silberberg [1], Magnus Claesson[1], Albert Suades[1], Lucie Delemotte [2], Göran Widmalm [3] ✉ & David Drew [1] ✉

In mammals, glucose transporters (GLUTs) mediate organism-wide sugar distribution, yet the molecular basis of substrate specificity remains unclear. The bacterial xylose transporter XylE serves as a model for GLUTs. However, although xylose and glucose bind with a similar affinity, xylose is transported, but glucose acts as an inhibitor. Here, using saturation transfer difference (STD) nuclear magnetic resonance (NMR) spectroscopy, we distinguished transported sugars from sugar inhibitors. Our findings revealed that only transported sugars generate STD NMR signals, which are abolished for xylose when XylE is trapped in either outward- or inward-facing conformations. Engineering the sugar-binding pocket and gating helix TM7b enabled glucose transport by XylE and corresponding STD signals. Using complementary molecular dynamics simulations, together with structural, biochemical and STD NMR analysis of related parasitic and mammalian GLUTs, we identified TM7b as a key determinant of occluded state formation. We conclude that, rather than the initial substrate-binding event observed in experimental structures, formation of a substrate-induced transition-state intermediate is the primary determinant of specificity in transporters.

Sugar porters belonging to the major facilitator superfamily (MFS) represent one of the largest groups of transporters across all kingdoms of life[1]. In humans, there are 14 different glucose transporters (GLUT), which differ in their kinetics, substrate preference and tissue localization[2,3]. GLUT1, for example, is required for uptake of glucose into the brain[4], whereas GLUT3 is required for the substrate translocation into neurons[4]. Other members, such as GLUT5 (refs. 5,6), do not transport glucose, but are instead required for the intestinal absorption of fructose. Incorrect expression and regulation of GLUTs is associated with a number of different diseases, in particular cancers and metabolic disorders[4,7]. Structures of GLUT1 (ref. 8), GLUT3 (ref. 9), GLUT4 (ref. 10), GLUT5 (ref. 11), GLUT7 (ref. 12), GLUT9 (ref. 13) and the *Escherichia coli* xylose transporter XylE[14–16] have been determined[17,18]. Together with

structures of the more distantly related sugar porters *Pf*HT1 from *Plasmodium falciparum*[19,20] and STP10 from *Arabidopsis thaliana*[21,22], we have a clear molecular basis for glucose coordination and the different conformations undergone during the GLUT cycle (Fig. 1a,b). Surprisingly, however, despite the wide range of D-glucose binding affinities[2] (from 0.007 to 10 mM), sugar coordination has been found to be remarkably well conserved in even the most sequence-divergent members[1,9,11,14,19–21] (Fig. 1b). The high structural conservation of the sugar binding site implies that there must be other differences outside this region that define substrate preferences and kinetics in GLUTs.

The hexose transporter *Pf*HT1 has evolved to transport a wide range of different sugars[3,23,24] and, as such, its structure has proven to be particularly informative for understanding the molecular

[1]Department of Biochemistry and Biophysics, Science for Life Laboratory, Stockholm University, Stockholm, Sweden. [2]Department of Applied Physics, Science for Life Laboratory, KTH Royal Institute of Technology, Stockholm, Sweden. [3]Department of Organic Chemistry, Stockholm University, Stockholm, Sweden. ✉e-mail: goran.widmalm@su.se; ddrew@dbb.su.se

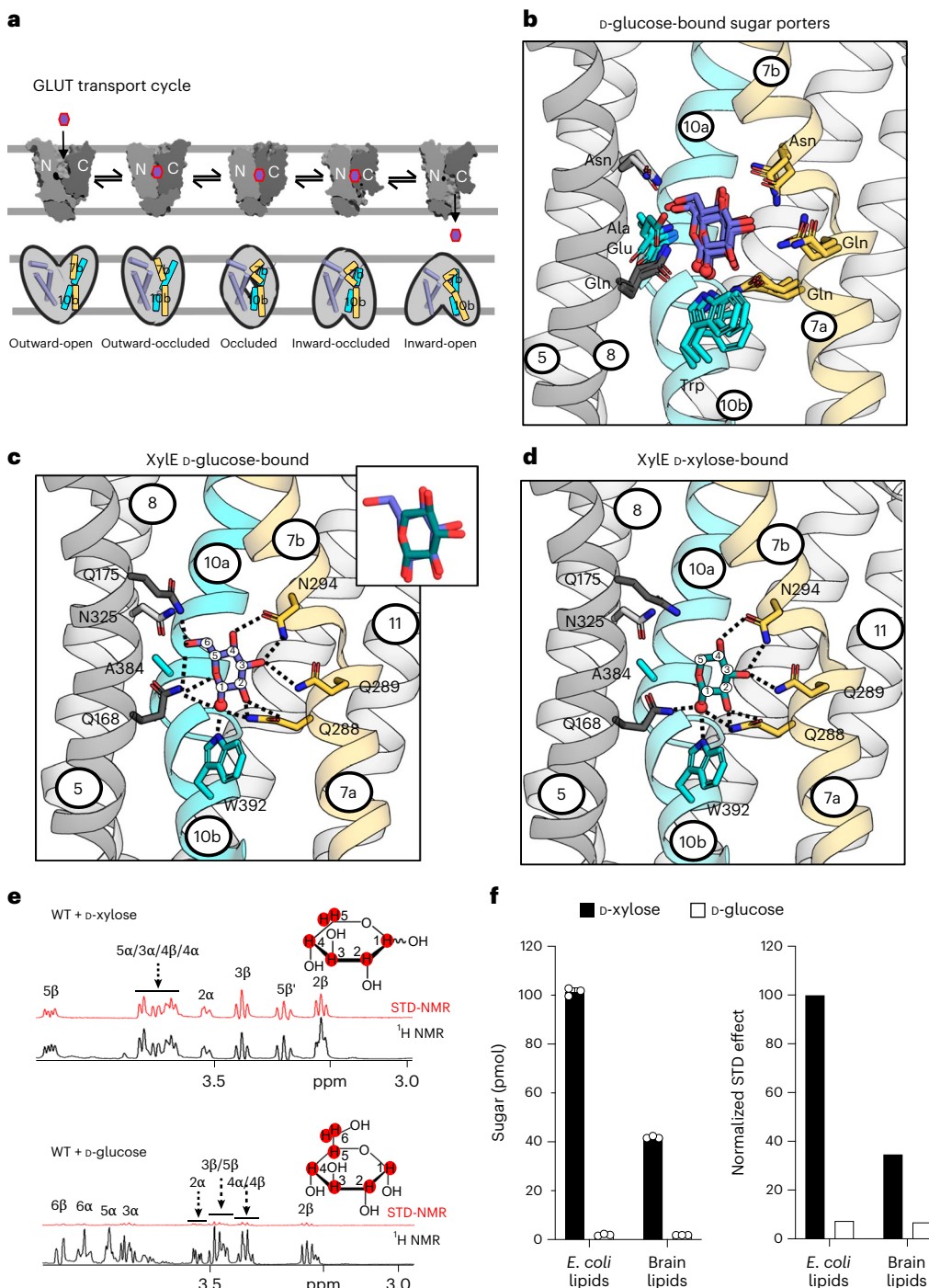

**Fig. 1 | GLUT transport cycle, sugar porter ligand binding and D-glucose and D-xylose interactions with XylE. a**, The sugar porter transporter cycle has six major conformations during the transition from outward-open to inward-open. The occluded state has been observed only in the malarial transporter *Pf*HT1 and is likely to be transient in GLUT proteins. **b**, Sugar binding site comparison between human GLUT3 (PDB 4ZWB), plant STP10 (PDB 7AAQ), *E. coli* XylE (PDB 4GBZ) and *P. falciparum Pf*HT1 (PDB 6RW3), with the respective bound carbohydrates shown as sticks, and their C1-hydroxyls accentuated as spheres for reference and TM7a-b and TM10a-b colored yellow and cyan, respectively, and protein as cartoon (TM1, TM2 and TM4 omitted for clarity). Although GLUT3 ($K_M$ = 1.3 mM)[28] and STP10 ($K_M$ = 0.007 mM)[50] transport D-glucose, *Pf*HT1 ($K_M$ = 0.9 mM)[28] can transport many different sugars in addition to D-glucose, whereas XylE cannot transport D-glucose[38], but binds it with the same affinity as its preferred substrate D-xylose. All residues hydrogen bonding to D-glucose (shown as sticks) are identical apart from a TM10a residue (Ala404

in *Pf*HT1), which in GLUT3 is a glutamate. Most residues in the N-terminal bundle surrounding the D-glucose, but not binding, are highly conserved (not shown). **c**, The sugar binding site and coordination of D-glucose in the outward-occluded crystal structure of XylE (PDB 4GBZ), shown as in **b**, and insert of the overlapping D-xylose and D-glucose (sticks, colored teal and slate, respectively) binding poses. **d**, The sugar binding site and coordination of D-xylose in the outward-occluded crystal structure of XylE (PDB 4GBY), where the interactions are very similar compared with those in **c**. Figure shown as in **b**. **e**, Top: D-xylose (500 μM) STD (red) and [1]H spectra (black) in presence of XylE−GFP reconstituted into liposomes. Bottom: as above for D-glucose. **f**, Left: normalized STD effects observed following addition of either D-xylose or D-glucose (500 μM) to XylE−GFP reconstituted in liposomes made from *E. coli* versus bovine brain-fraction-seven lipids. Right: uptake of [3]H-D-xylose or [14]C-D-glucose by WT XylE−GFP in proteoliposomes. Error bars: mean ± s.e.m. of *n* = 3 independent experiments.

determinants for sugar specificity[19]. More specifically, *Pf*HT1 has acquired the ability to transport both D-glucose and D-fructose sugars with similar kinetics ($K_M$) as the dedicated high-affinity D-glucose (GLUT3) and D-fructose (GLUT5) transporters, respectively[3,23,24]. Although the coordination of D-glucose in *Pf*HT1 was found to be almost identical to that of GLUT3 (refs. 19,20) Fig. 1b, the occluded structure of *Pf*HT1 revealed that the extracellular gate TM7b that connects D-glucose binding with outside-facing occlusion closes off the sugar-binding pocket completely (Fig. 1a). Principle component analysis[19] and molecular dynamic (MD) simulations[25] confirmed that this state represents an occluded conformation. Although this state is expected to be unstable in GLUT5 and other GLUT proteins, the *Pf*HT1 protein has an unusual extracellular TM7b gate that is more polar[19]. Pointedly, two highly conserved tyrosine residues in TM7b of GLUT proteins are replaced in *Pf*HT1 with the polar residues serine and asparagine[19] and MD simulations show that this difference enables the extracellular gate to close spontaneously more easily[19]. *Pf*HT1 mutations of TM7b gating residues and residues in TM1 that interacted with TM7b were found to be just as critical to transport as residues coordinating D-glucose directly[19]. Interestingly, the mutation of several TM7b gating residues shifted the substrate preference of *Pf*HT1 from D-glucose towards D-fructose[19]. It seems that *Pf*HT1 has not evolved the sugar binding site to transport many sugars, but instead adapted its interplay with the TM7b gate. Simplistically, *Pf*HT1 is thought to be less selective in which sugars are transported because TM7b shuts more easily. This allostery-driven transport model is consistent with forward-evolution screens of hexose transporters in yeast, wherein single-point mutations in the TM7b gate were uncovered that shifted the preference of transporting D-glucose towards to D-xylose[26,27]. In contrast, no single-point mutations in the sugar-binding site with shifted substrate specificity were uncovered.

Although the importance of the extracellular gate TM7b seems clear, a key open question remaining is the coupling between TM7b and the binding and transport of different sugars. To answer this question, we need to be able to assess sugar binding as well as transport, but this has not been routinely possible for GLUTs due to the (weak) affinities for sugars, which are in the millimolar range. It is well known that lipids are also critical for GLUT transport function[28,29] and thus we would ideally want to assess binding in a membrane bilayer. Here, using purified XylE, *Pf*HT1 and mammalian GLUT5 sugar transporters reconstituted into liposomes, we have probed the interaction of different sugars by saturation transfer difference (STD) nuclear magnetic resonance (NMR) at room temperature. Furthermore, we carried out MD simulations and determined the crystal and cryo-EM structures of *Pf*HT1 in complex with the fructose analog 2,5-anhydro-D-mannitol (2,5-AHM). Our combined data support a two-step model for transport catalysis wherein, upon conformational selection of the sugar, the extracellular gate TM7b becomes stabilized in an outward-occluded state that increases the probability of the sugar catalyzing an induced-fit to the occluded state and, in doing so, promotes bundle isomerization and transport.

## Results and discussion

### Establishing STD NMR for probing sugar binding to XylE proteoliposomes

The *E. coli* XylE transporter has evolved to transport D-xylose, whereas D-glucose acts as a competitive inhibitor[30]. The reason for D-glucose inhibition is unclear as, according to isothermal-titration calorimetry (ITC), the binding affinities for D-xylose and D-glucose are similar at 0.35 mM and 0.77 mM, respectively[14,31]. Further, X-ray structures show that both sugars stabilize a substrate-induced outward-occluded conformation[14] (Fig. 1c,d). In fact, the difference between substrates and inhibitors poses a fundamental question in transport biology in general[32] and probably involves a comprehension of how occluded states are formed. Unfortunately, this state is particularly difficult to capture experimentally and is thus poorly understood.

To assess sugar binding to XylE in liposomes, we decided to use STD NMR, which has been used previously to examine how ligands interact with membrane proteins in dynamic environments[32,33]. Essentially, substrate protons that are in close proximity with the protein will receive a higher degree of saturation and thus produce STD NMR signals, whereas protons that interact only transiently will not give rise to STD NMR signals[34]. As only the substrate protons are monitored[35], no labeling of the protein is required and this NMR technique is also compatible with lipid mimetics. Despite these strengths, STD NMR has been used only a few times to monitor ligand–membrane protein interactions in liposomes[32,36]. However, monitoring sugar–protein interactions in a membrane bilayer rather than in detergent is crucial, as the lipid composition has a large impact on sugar transport rates[28] and is therefore likely to influence the energetic barriers between outward-facing and inward-facing states[37]. Using the rapid-dilution method we have found that the protein reconstitution efficiency of sugar transporters into liposomes remains similar between different preparations[37] and by working with purified green fluorescent protein (GFP) fusions we further ensured the final substrate-to-protein ratio was kept constant by adjusting the amount of proteoliposomes based on GFP fluorescence (Methods). For similar reasons, GFP fusions were also used for all transport activity measurements.

After optimization, the STD spectrum was of sufficient intensity to resolve most of the nondeuterated hydrogens of D-xylose binding to XylE incorporated into liposomes (Fig. 1e, Supplementary Table 1, Extended Data Fig. 1a and Methods). From these spectra, we were able to calculate the degree of saturation transfer from protein to sugar protons. As the degree of saturation transfer is proportional to the distance of the sugar proton(s) to the protein, we could establish that most protons in D-xylose contribute to the binding to XylE (Fig. 1e,f). Unexpectedly, STD NMR signals for D-glucose addition to XylE were very low (Fig. 1e,f and Supplementary Table 1). The low intensity STD NMR signal for D-glucose indicated that the method might produce signals specific to transported sugars. Consistent with this rationale, we observed higher STD NMR signals for D-xylose when XylE was reconstituted into liposomes made from *E. coli* lipids rather than brain lipids, in agreement with the observation that XylE has higher transport activity in the *E. coli*-specific lipid composition[28] (Fig. 1f and Supplementary Table 1). We then confirmed that empty liposomes produced no STD NMR signals and that the addition of 50-fold excess D-glucose abolished the STD signal for D-xylose when added to the same proteoliposome preparation (Extended Data Fig. 1b,c).

We hypothesized that perhaps only transported sugars gave rise to STD NMR signals. To test our hypothesis, we used mutagenesis to lock XylE in either the outward- or the inward-facing conformations. For outward-facing locking, residues L315 and G58 were substituted to tryptophan (XylE-WW), as this variant was shown previously to stabilize an outward-occluded conformation[31] (Fig. 2a, Extended Data Fig. 1a and Supplementary Table 1). Analysis by ITC had shown that the XylE-WW variant retained D-xylose binding with an affinity around fourfold higher than that of wild type (WT)[31] ($K_d = 0.08$ mM). To lock the inward-facing state, residues V35 and E302 were substituted with cysteines (XylE-CC), enabling the formation of a disulfide bond under oxidizing conditions (Fig. 2b and Extended Data Fig. 1a). In the XylE-CC variant, D-xylose binds with an affinity ($K_d = 0.63$ mM) around twofold lower than that of WT[31]. As expected, purified XylE-WW was well-folded and showed no uptake of $^3$H-D-xylose, whereas the XylE-CC variant showed transport activity only under reducing conditions (Fig. 2a,b, Extended Data Fig. 1a and Supplementary Table 1). We repeated the STD NMR analysis on the XylE-WW and XylE-CC (oxidized) and XylE-CC (reduced) variants. As shown in Fig. 2a,b and Supplementary Table 1 the D-xylose STD NMR signals for XylE-WW were now almost completely abolished and the XylE-CC (oxidized) variant showed a 75% reduction of the WT STD NMR signal; we interpret the remaining response as the inability to either fully form or retain disulfide-trapped XylE

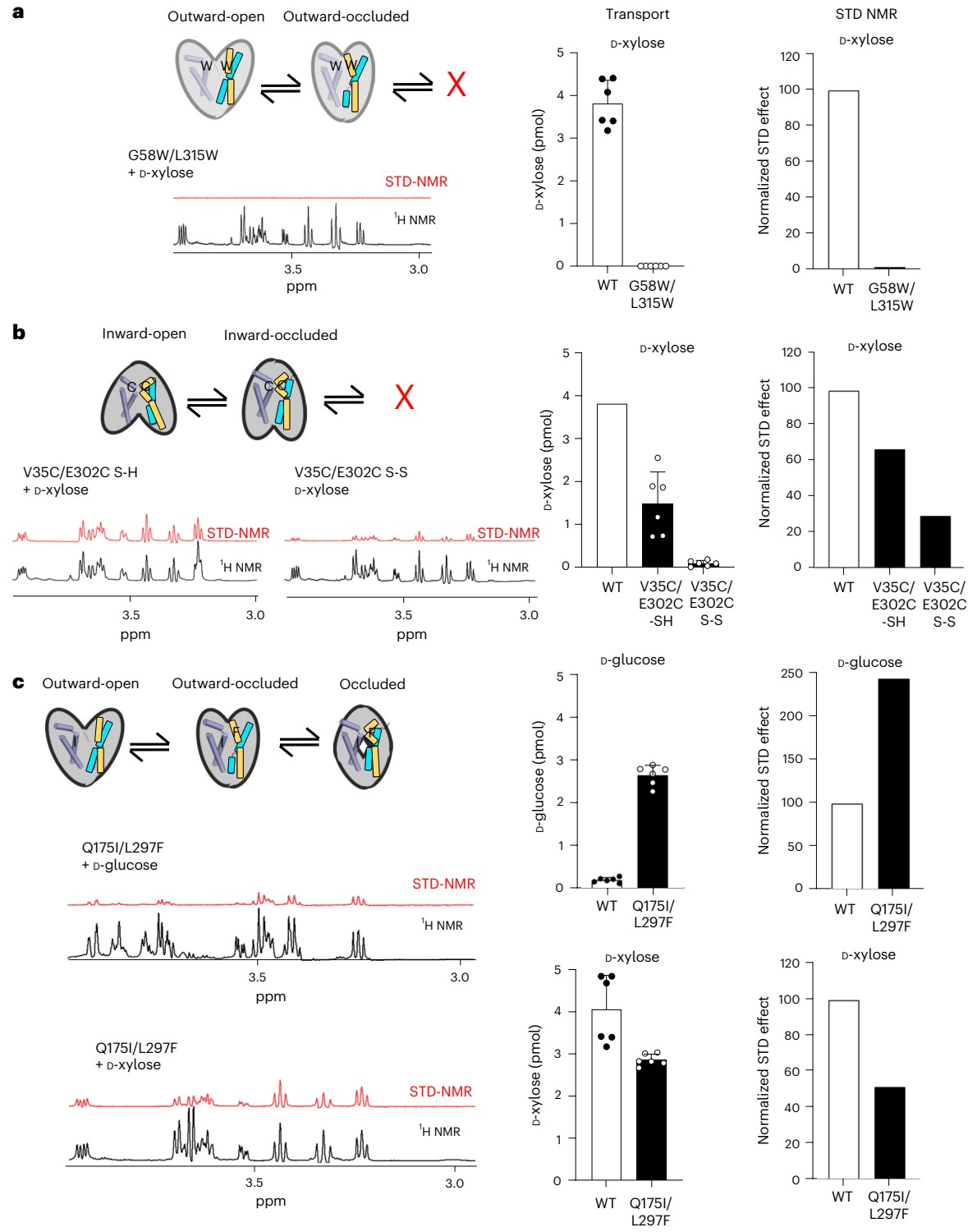

**Fig. 2 | Only transported sugars produce STD NMR signals. a**, Left: schematic of transporter states for the XylE double tryptophan mutant locked in the outward-facing state and ¹H and STD spectra. Right: transport data and normalized STD effects. Error bars: mean ± s.e.m. of *n* = 6 independent experiments. **b**, Left: schematic of transporter states for XylE double cysteine mutant locked in inward-facing state under oxidizing conditions with ¹H and STD spectra. Right: transport and normalized STD effects per mutant under reducing and oxidizing conditions. WT data are reused as reference from **a**. **c**, Left: schematic of first half of the transport cycle for the XylE mutant Q175I and L297F capable of transporting both D-glucose or D-xylose with respective ¹H and STD spectra for D-glucose (middle) and D-xylose (bottom). Right: transport and normalized STD effects for D-glucose (top) and D-xylose (below). Error bars: mean ± s.e.m. of *n* = 6 independent experiments.

over the 18-h measuring timeframe of the NMR experiments at room temperature (Methods).

Based on differences between a GLUT1 homology model and the XylE crystal structure, Q175I and L297F variants in XylE were identified

previously as enabling weak D-glucose transport[38]. The TM5 residue Q175 in XylE forms an additional hydrogen bond to the C6–OH of D-glucose and in GLUTs the corresponding residues is an isoleucine (Fig. 1c). Consistent with previous analysis, the Q175I variant shows a

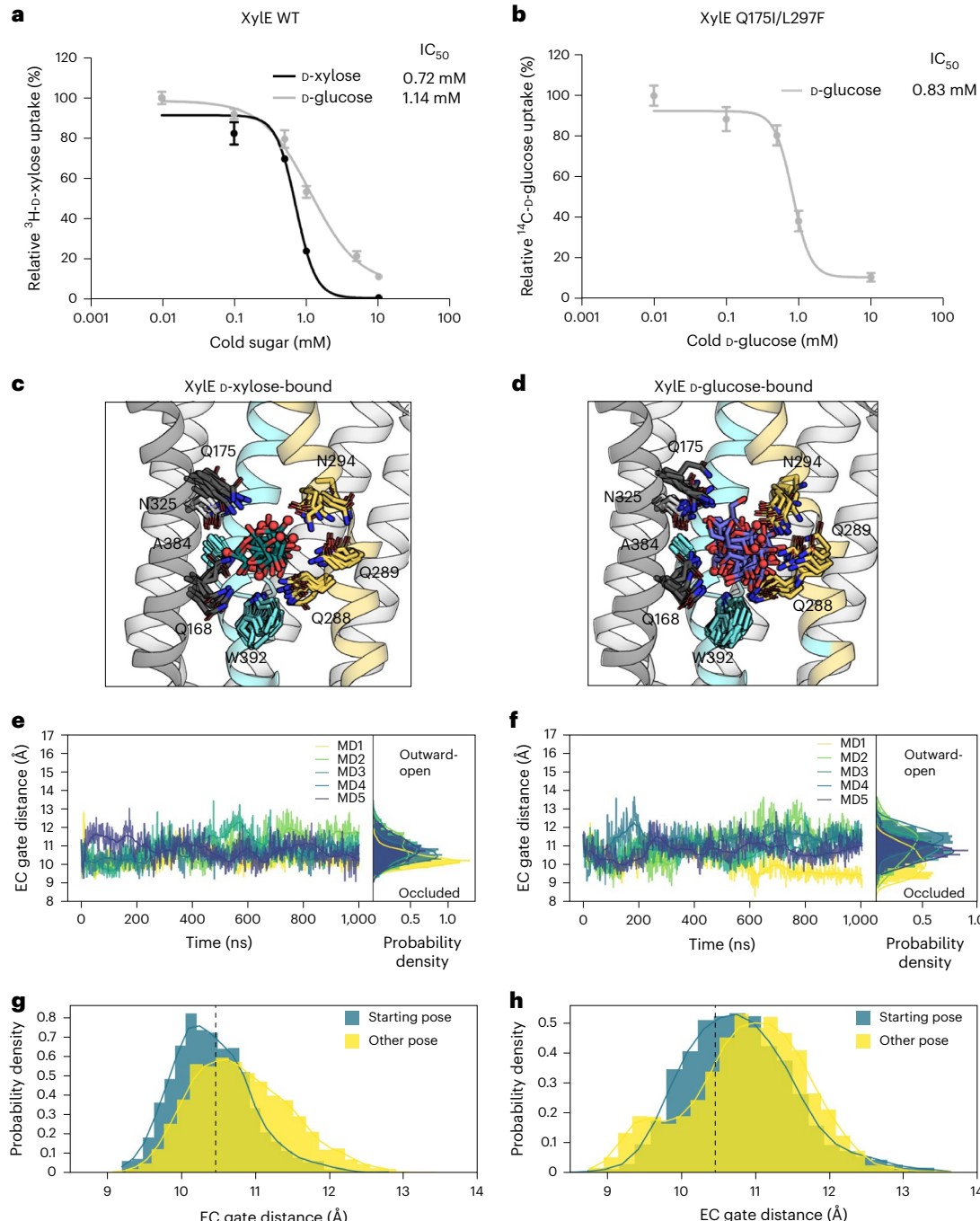

**Fig. 3 | XylE IC$_{50}$ values and protein–sugar interactions probed by MD simulations. a**, IC$_{50}$ values for XylE–WT uptake of $^3$H-D-xylose in the presence of either cold D-xylose or D-glucose. Error bars: mean ± s.e.m. of $n$ = 6 independent experiments. **b**, IC$_{50}$ values for XylE double mutant XylE-Q175I/L297F uptake of $^{14}$C-D-glucose in the presence of cold D-glucose. Error bars: mean ± s.e.m. of $n$ = 6 independent experiments **c**, D-Xylose (teal, sticks) and coordinating residues (sticks) in the frequently populated clusters (for example, with more than two members) for five independent 1-μs-long simulations using the D-xylose complex structure (helices shown as cartoon with TM1, TM2 and TM4 omitted

for clarity colored as in Fig. 1b). The O1 atom of D-xylose is shown accentuated by red spheres for reference. **d**, D-Glucose (blue, sticks) and coordinating residues (sticks) in the frequently populated clusters in five 1-μs-long simulations of the D-glucose-bound structure shown as in **c**. **e,f**, Extracellular gate distances over time and the resulting probability density for each of the five repeats for D-xylose (**e**) and D-glucose (**f**). **g**, Comparison between the probability density of extracellular gate distances with D-xylose for frames in which the sugar occupies the pose resolved in the published structure (blue) and any other pose (yellow). **h**, As in **g** but for D-glucose.

WT-like D-xylose transport, yet was still unable to transport D-glucose (Extended Data Fig. 1d). The residue L297 was previously modeled to be located peripheral to the sugar-binding site[38], but the corresponding residue has since shown to be positioned in the gating helix TM7b[1]. In GLUT1–4 members, L297 corresponds to a phenylalanine, and recent coevolution analysis predicts an encouraged contact

between this TM7b residue and TM1 in forming the occluded conformation[39]. The mutation of L297 to phenylalanine abolished D-xylose transport and likewise showed no ability to transport D-glucose (Extended Data Fig. 1d). In contrast to single-point mutations, combining the Q175I and L297F mutations enabled D-glucose transport in our optimized proteoliposome assay[28], almost as efficiently as the

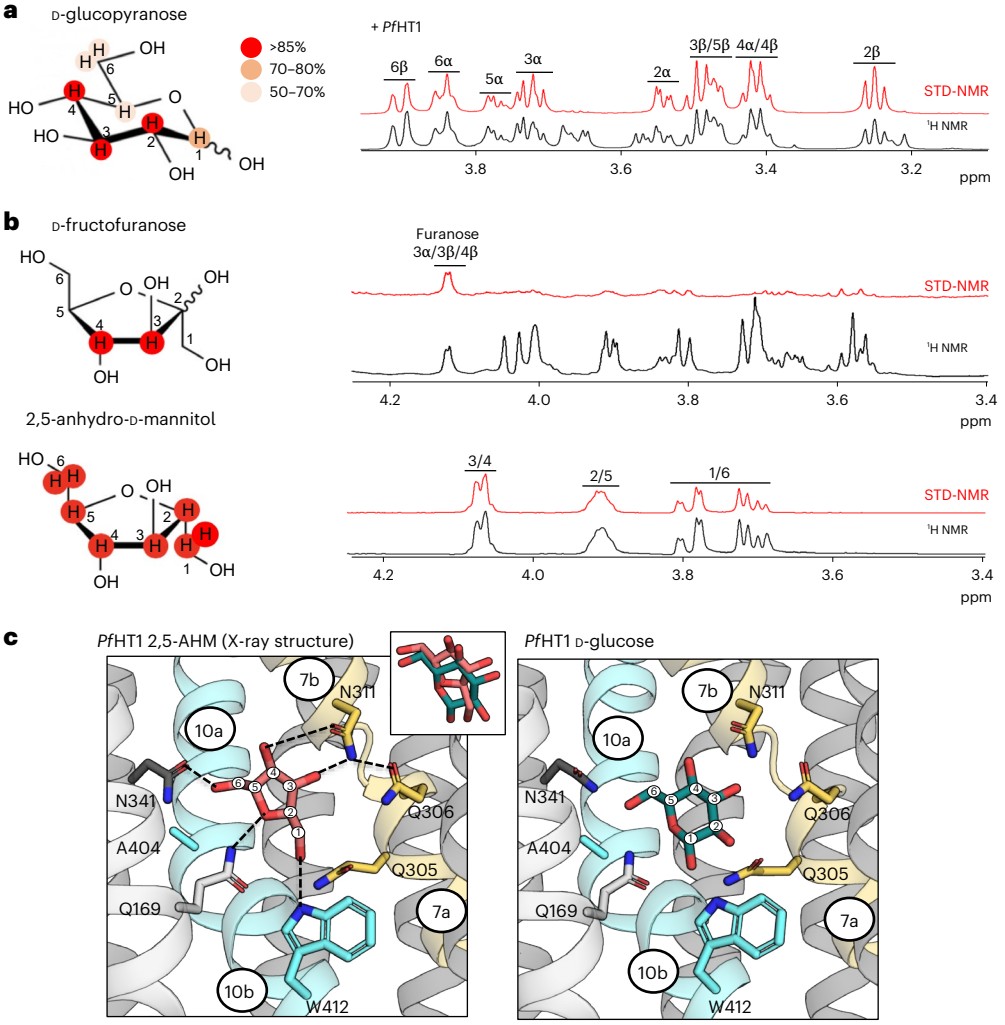

**Fig. 4 | D-Glucose versus 2,5-AHM coupling in *Pf*HT1. a**, Left: STD-derived epitope mapping of the *Pf*HT1:D-glucose proton interaction degree is shown on the structure of D-glucose by color from the highest (red) to lowest (beige). The STD/off-resonance ratio of the H2 β-proton, which was the highest value of all protons, was set as 100% for the normalization of the other proton resonance intensities. Right: STD (red) and ¹H spectra (black) of D-glucose (500 µM) in the presence of *Pf*HT1-GFP reconstituted into liposomes. The spectral region of the H1 proton is not shown for clarity. **b**, Left: structures of D-fructofuranose (top) and 2,5-AHM (bottom) with protons producing STD signals shown on red background. Right: corresponding STD (red) and ¹H spectra (black) in the presence of *Pf*HT1-GFP reconstituted into liposomes. **c**, Sugar binding site in the crystal structure of *Pf*HT1 in complex with the sugar 2,5-AHM (left) and in comparison the coordination of D-glucose (right), determined previously (PDB 6RW3); sugar binding site residues and ligands are shown as sticks, hydrogen bond interactions indicated by dashed lines and protein as cartoon (as in Fig. 1b). Insert shows the overlapping binding position of these carbohydrates following superimposition of the two structures.

WT levels for D-xylose (Fig. 2c). Consistently, STD NMR signals were now detectable for both D-glucose and D-xylose sugars, respectively (Fig. 2c and Supplementary Table 1). Thus, we can conclude that STD NMR signals are detected only for sugar–protein pairings culminating in substrate transport. Our analysis further confirms the importance of the allosteric coupling between sugar binding and TM7b gating.

Although the affinity of XylE for D-glucose and D-xylose were reported to be similar by both ITC[31] and solid-supported-membrane-based electrophysiology[40], the absence of STD NMR signals from D-glucose could also be explained with a much higher binding affinity for the tested substrate–transporter pairings. Albeit unlikely, we resolved to rule out this interpretation. We reasoned that the most straightforward analysis would be to measure the comparative competition of ³H-D-xylose uptake by externally added cold D-xylose versus D-glucose sugars. As shown in Fig. 3a, we could confirm that D-glucose was clearly no better at competing for ³H-D-xylose uptake than unlabeled D-xylose, with estimated half-maximal inhibitory concentration ($IC_{50}$) values of 1.1 and 0.73 mM, respectively. We likewise confirmed that competition of D-glucose for

¹⁴C-D-glucose uptake by the Q175I/L297F variant was similar to WT with an $IC_{50}$ of 0.83 mM (Fig. 3b).

To explain why only sugar–protein pairings enabling transport produce STD NMR signals, we carried out equilibrium-based MD simulations of the D-xylose and D-glucose-bound outward-occluded WT structures (Methods). Despite extensive hydrogen bond interactions between the sugar and binding site residues, both D-xylose and D-glucose sugars were, on average, highly mobile in the sugar-binding pocket (Fig. 3c,d). Across 5 × 1-µs MD simulations of the D-xylose-bound XylE crystal structure[14], in only one of the five repeated MD simulations (MD4) was D-xylose bound stably for most of the simulation time (823 out of 1,000 total frames; Fig. 3e and Supplementary Videos 1 and 2). In contrast, D-glucose was more mobile and the longest stably bound state was observed for only ~60% of the simulation time (Fig. 3f and Supplementary Video 3). Following TM7b gating by the degree of extracellular gate closure we found that, in two simulations where D-xylose was bound stably for a longer period of time (MD1 and MD4), TM7b was retained in a closed state (Fig. 3e and Extended Data Fig. 2a,b).

Alternatively, in one of the MD simulations (MD5) the D-xylose sugar exited quickly (Extended Data Fig. 2a). Essentially, we observed a coupling between the D-xylose being bound for a longer time in the starting pose (that is, the position in the structure) and the occlusion of the extracellular gate (Fig. 3g). In contrast, in the 5 × 1-μs MD simulations of the D-glucose-bound XylE structure[14], the TM7b gate for D-glucose retained an outward-occluded conformation, whereas the degree of tumbling showed no clear correlation with the gate closure (Fig. 3f,h and Extended Data Fig. 2c,d).

MD simulations suggest that, although D-xylose is sometimes able to shift the population towards an occluded state, D-glucose was able to retain only an outward-occluded conformation, consistent with a possible explanation of its inhibitory mode of action. Similar results are obtained in MD simulations of outward-occluded GLUT3, where the protein also showed a flexible TM7b gate and the sugar similarly tumbles around its in crystallo observed position (Extended Data Fig. 3a–d). Similar to XylE with D-xylose, the most closed position of TM7b in human GLUT3 was associated with the longest stable coordination of D-glucose (Extended Data Fig. 3b). Rather than different substrate affinities, we conclude that poor STD NMR signals are the result of glucose failing to stabilize an occluded conformation in XylE, hence not allowing a long enough interaction time with coordinating residues for sugar protons to achieve sufficient magnetization transfer. Thus, we suggest that the difference between transported sugars and sugar inhibitors lies in their ability to conformationally stabilize and induce a transition to the occluded state, thus obtaining the strongest coordination for the sugar and, thereby, receiving the saturation required for producing STD NMR signals.

### Sugar coupling between D-glucose and D-fructose sugars

Having concluded that STD NMR can detect only the interaction with transported substrates, we moved our attention to *Pf*HT1, trying to dissect and learn from its ability to transport many different sugars. Similar to D-xylose binding to XylE, the STD NMR spectrum for D-glucose binding to *Pf*HT1 in liposomes was allowed us to distinguish all the nondeuterated hydrogens in D-glucose individually, except for proton peaks overlapping with the solvent peak (Fig. 4a and Extended Data Fig. 4a,b). The epitope mapping shows that the H2, H3 and H4 protons interact more with *Pf*HT1 than the H1, H5 and H6 protons. This analysis is consistent with the observation that the –OH group orientation in the C3 and C4 position is the most critical for D-glucose coordination[2,19,23]. The STD NMR analysis is further consistent with the hydrogen bonding distance between the hydroxyl protons in D-glucose and the sugar coordinating side chains. Thus, STD NMR analysis supports the side-chain coordination of D-glucose seen in crystal structures of *Pf*HT1 determined under cryogenic conditions in the occluded state.

We continued the STD NMR analysis of *Pf*HT1 with D-fructose. Although *Pf*HT1 in the presence of D-fructose showed the same degree of saturation transfer as in the presence of D-glucose, we could observe STD signals arising only from the C3 and C4 protons (Fig. 4b and Supplementary Table 2). The STD NMR spectra produced exclusively the ¹H NMR for D-fructofuranose, which comprises only ~29% of the equilibrium mixtures of possible D-fructose tautomers[41]. This observation is consistent with transport assays that *Pf*HT1 prefers the furanose form of D-fructose[24]. To assess whether sugar porters prefer D-fructofuranose in general, we repeated the STD NMR analysis for *rat* GLUT5 embedded into liposomes (Extended Data Fig. 4c and Supplementary Table 2). Like *Pf*HT1, proton spectra were observed only from the D-fructofuranose sugar.

To provide additional evidence for the preference of D-fructofuranose, we remeasured binding to *Pf*HT1 with the D-fructose analog 2,5-AHM, which is almost identical to D-fructose, but is devoid of the C2-hydroxyl group, and exists only in the furanose form (Fig. 4b). STD NMR experiments showed improved signals of 2,5-AHM to *Pf*HT1 as compared to D-fructose, with difference spectra now apparent for

### Table 1 | X-ray data collection and refinement statistics

| | *Pf*HT1_2,5-AHM (PDB 9HKK) |
|---|---|
| **Data collection** | |
| Space group | P 1 21 1 |
| Cell dimensions | |
| a, b, c (Å) | 123.92, 70.75, 181.36 |
| α, β, γ (°) | 90, 107.06, 90 |
| Resolution (Å) | 25.33–3.55 (3.68–3.55)ᵃ |
| $R_{merge}$ | 0.3355 (3.016) |
| Mean I / σI | 5.02 (0.73) |
| CC1/2 | 0.976 (0.276) |
| CC* | 0.994 (0.658) |
| Completeness (%) | 98.69 (98.20) |
| Redundancy | 6.5 (6.8) |
| **Refinement** | |
| Resolution (Å) | 25.33–3.55 |
| No. of reflections | 36,417 |
| $R_{work}$/$R_{free}$ | 0.234/0.276 |
| No. of atoms | |
| Protein | 15,321 |
| Ligand/ion | 806 |
| Water | 20 |
| B factors (Å²) | |
| Protein | 138.6 |
| Ligand/ion | 129.6 |
| Water | 72.7 |
| R.m.s. deviations | |
| Bond lengths (Å) | 0.003 |
| Bond angles (°) | 0.69 |
| Ramachandran plot | |
| Favored (%) | 96.6 |
| Allowed (%) | 3.21 |
| Disallowed (%) | 0.21 |

ᵃValues in parentheses are for highest-resolution shell.

all nondeuterated hydrogens (Fig. 4b and Supplementary Table 2). The H3 and H4 protons gave the largest difference spectra and suggests that 2,5-AHM and D-fructose will be coordinated in a similar orientation to D-glucose. Indeed, biochemical analyses indicate D-fructose will be transported by GLUT5 with the C1–OH group facing the endofacial direction as in D-glucose, as substituents to D-fructose were better tolerated when added to the C6–OH position[42,43].

Until now, structures of GLUTs have been obtained only in complex with one substrate sugar, mostly glucose[1,9,11,14,19–21]. Guided by the STD NMR analysis, we were able to determine the crystal structure of *Pf*HT1 in complex with 2,5-AHM to 3.5 Å resolution (Table 1 and Fig. 4c). Similar to the D-glucose-bound *Pf*HT1 structure[19], *Pf*HT1 crystallized as a dimer with four molecules present in the asymmetric unit. The *Pf*HT1 structure was obtained in an overall similar conformation to that seen previously. The sugar coordinating residues were positioned similarly to the D-glucose-bound structure, but now there was additional electron density matching 2,5-AHM (Extended Data Fig. 4d). The C3- and C4-hydroxyl groups of 2,5-AHM formed hydrogen bonds to N311 residue as in the D-glucose complex, whereas the ring oxygen

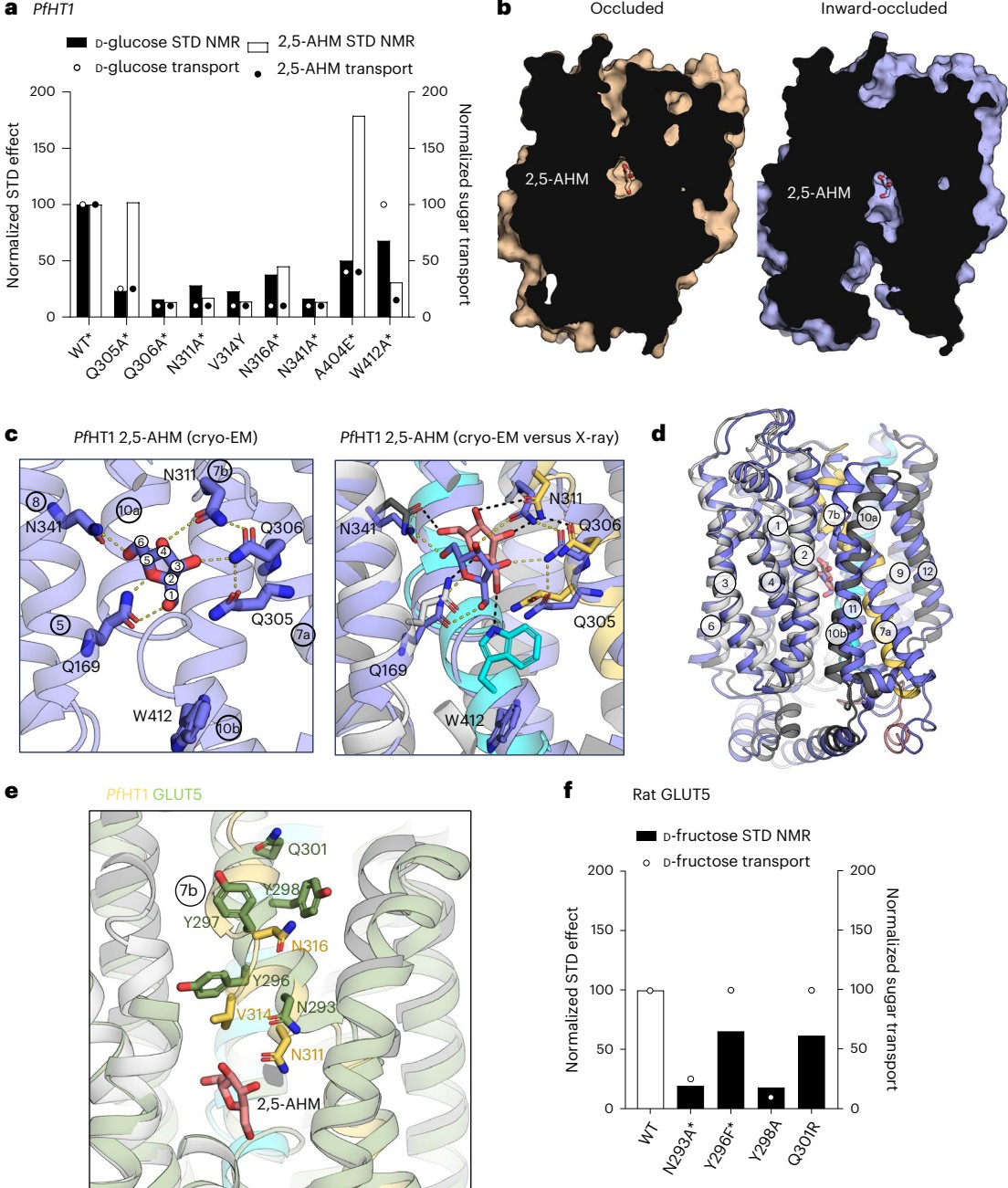

**Fig. 5 | Sugar-coupling analysis of *Pf*HT1 and rat GLUT5. a**, Normalized STD effects and transport activity of sugar binding residue mutants of *Pf*HT1 in the presence of D-glucose or 2,5-AHM. Bars: STD effects; circles: percentage of sugar transport compared to WT for D-glucose and 2,5-AHM, with previously determined values (asterisk, as reported in ref. 19). **b**, Left: surface representation cross-section of the 2,5-AHM bound fully occluded (wheat) X-ray structure of *Pf*HT1. Right: surface representation cross-section of the 2,5-AHM bound inward-occluded (blue) structure of *Pf*HT1 obtained by cryo-EM. Ligand shown as sticks. **c**, Left: binding site in the cryo-EM structure of *Pf*HT1 in complex with 2,5-AHM (blue sticks) in the inward-occluded conformation. Right: comparison of the 2,5-AHM coordination between the fully occluded *Pf*HT1 X-ray structure (2,5-AHM as

pink sticks, interacting residues and helices as cyan, yellow and gray) to inward-occluded cryo-EM complex (shown as on left). Hydrogen bond interactions are indicated by dashed lines. **d**, Overall structural comparison of the *Pf*HT1 2,5-AHM complexes in the fully occluded and inward-occluded states colored as in **c**. Upon inner gate open/closing the substrate would be displaced vertically by ~2 Å. **e**, The gating residues in the 2,5-AHM X-ray complex structure of *Pf*HT1 (yellow sticks, with protein and ligand shown as in **c**) compared to GLUT5. Right: comparison with rat GLUT5 (green) (PDB 4YBQ). **f**, Normalized STD effects for WT and mutants of rat GLUT5 in the presence of D-fructose; bars: STD effect values, circles: percentage of sugar transport compared to WT, with values determined previously (asterisks) taken from ref. 25 or as shown in Extended Data Fig. 7b.

and the C1–OH group hydrogen bonded to Q169 and W412, respectively (Fig. 4c). With the absence of an OH group at the C2 position, the residue Q305 was not required for hydrogen bond formation with 2,5-AHM; the Q305 residue is, however, required for D-glucose and D-fructose coordination (Fig. 4c). Nevertheless, as 2-deoxyglucose and D-glucose have similar kinetics in GLUT1 and GLUT3 transporters[2], coordination

at the C2–OH position could be considered flexible. The residue W412 is the only side chain forming a hydrogen bond interaction to the C1–OH group of 2,5-AHM and the structure thus explains why its substitution to alanine selectively abolished D-fructose transport[19]. In contrast, in an W412A variant, D-glucose could still be coordinated at the C1–OH position by Q169 and Q305 residues (Fig. 4c).

## Table 2 | Cryo-EM data collection, refinement and validation statistics

| | PfHT1_2,5-AHM (EMDB-54787) (PDB 9SDL) |
|---|---|
| **Data collection and processing** | |
| Magnification | 130,000 |
| Voltage (kV) | 300 |
| Electron exposure (e⁻/Å²) | 62.3 |
| Defocus range (μm) | 2.0–0.6 |
| Pixel size (Å) | 0.65 |
| Symmetry imposed | C2 |
| Initial particle images (no.) | 5,741,311 |
| Final particle images (no.) | 300,678 |
| Map resolution (Å) | 2.42 |
| FSC threshold | 0.143 |
| Map resolution range (Å) | 2.1–4.2 |
| **Refinement** | |
| Initial model used (PDB code) | 6RW3 |
| Model resolution (Å) | 2.6 |
| FSC threshold | 0.5 |
| Model composition | |
| Non-hydrogen atoms | 7,604 |
| Protein residues | 956 |
| Ligands | 2 |
| *B* factors (Å²) | |
| Protein | 76.72 |
| Ligand | 66.07 |
| R.m.s. deviations | |
| Bond lengths (Å) | 0.003 |
| Bond angles (°) | 0.430 |
| Validation | |
| MolProbity score | 1.59 |
| Clashscore | 2.59 |
| Poor rotamers (%) | 0.47 |
| Ramachandran plot | |
| Favored (%) | 97.48 |
| Allowed (%) | 2.52 |
| Disallowed (%) | 0.0 |

To gain a deeper understanding of the D-glucose versus D-fructose recognition by *Pf*HT1, we analyzed the STD NMR signals for several sugar-binding site variants (Extended Data Figs. 4a,b and 5a). Alanine substitutions of the sugar coordinating residues Q306, N311 and N341 all showed low intensity STD NMR signals (Fig. 5a and Supplementary Table 2), consistent with transport assays that have demonstrated that these variants abolished D-glucose and D-fructose transport[19]. STD NMR signals of the Q305A variant showed WT-like binding for 2,5-AHM, although this mutant showed poor D-fructose transport[19]. However, the crystal structure of 2,5-AHM is consistent with this STD NMR analysis, as the Q305 residue does not form an interaction with the sugar 2,5-AHM (Fig. 4c). Previously, the *Pf*HT1 residue A404 was mutated to glutamate to match the glucose-specific isoform GLUT3 (ref. 9), and the tryptophan residue W412 was mutated to alanine to match the

fructose-specific isoform GLUT5 (ref. 11). Contrary to expectation, but consistent with proteoliposome assays and the crystal structure[19], the W412A variant diminished STD NMR signals for 2,5-AHM more than for D-glucose, whereas the A404E variant showed diminished STD NMR signals only for D-glucose (Fig. 5a and Supplementary Table 2). Taken together, the STD NMR analysis is consistent with the transport analysis of sugar binding site mutations and the coordination of D-fructose and D-glucose sugars observed in the crystal structures.

### Cryo-EM structure of *Pf*HT1 with 2,5-AHM provides structural evidence for two-step mechanism of sugar transport

Until now, only the plant sugar porter STP10 has been captured in more than one substrate-bound conformation[44] (Extended Data Fig. 5b). Using cryo electron microscopy (cryo-EM), we further determined the structure of *Pf*HT1 in MSP1E2 nanodiscs—using synthetic lipids retaining transport activity[28]—in complex with 2,5-AHM to an improved overall resolution of 2.4 Å (Extended Data Fig. 6a and Table 2). Structure determination was facilitated by the retention of a stable *Pf*HT1 homodimer, in which the dimer interface was formed by interactions between the N-terminal bundles and also contained well-defined densities for lipids, Extended Data Fig. 6a. Under these conditions, the dominant population of *Pf*HT1 is an inward-occluded conformation (Fig. 5b). Map density unambiguously supported the modeled position of 2,5-AHM, consistent with the occluded *Pf*HT1 crystal structure (Fig. 5c and Extended Data Fig. 6b,c). Strikingly, although the hydrogen bonding geometry of N311, N341 and Q169 is essentially unchanged relative to the occluded state, the residue W412 in TM10b is positioned ~7 Å from the C1–OH group and does not interact with the substrate (Fig. 5c). In the previously described occluded structure, W412 hydrogen bonds to the C1–OH. Thus, although TM10b has rotated inwards, it has not completed the conformational closure to that observed in the fully occluded *Pf*HT1 crystal structure (Fig. 5c).

Because this coordination was unexpected, we re-examined all available substrate-bound sugar-porter structures. Few such structures exist and, aside from occluded *Pf*HT1, most also have been obtained with an additional lipid or detergent molecule interacting with the TM7b or TM10b gating helices (Extended Data Fig. 5b). A notable exception is the recent inward-occluded cryo-EM structure of human GLUT9 with urate bound[13,45] that was also determined without additional coligands, and the position of urate parallels closely that of 2,5-AHM in *Pf*HT1 (Extended Data Fig. 6d). In both cases, the sugar is coordinated only partially in the inward-occluded state. Likewise, the binding pose of 2,5-AHM resembles closely that of the glucoside in the GLUT1 crystal structure with n-nonyl-β-D-glucopyranoside (β-NG; Extended Data Fig. 6d), where the detergent tail probably prevented full inward closure of TM10b and thereby hinders the glucoside moiety from interacting with the tryptophan equivalent to W412 (ref. 8). Taken together, structural comparisons, STD NMR analysis and transport assays confirming the essential role of W412 in 2,5-AHM recognition, indicates that the sugar occupies a suboptimal binding pose in the inward-occluded state of *Pf*HT1.

A detailed comparison of the inward-occluded and fully occluded structures shows that 2,5-AHM is displaced ~2 Å vertically (Fig. 5c,d and Supplementary Videos 4 and 5). In the occluded state, interaction with all hydroxyl groups of 2,5-AHM is accompanied by a contraction of the sugar-binding pocket relative to the inward-occluded conformation (Fig. 5b). Together with the STD NMR analysis, the inward-occluded cryo-EM structure of *Pf*HT1 suggests that partially occluded states are only long-lived initial modes of sugar coordination—stabilized either by low temperatures or by additional interactions of coligands involving the gating helices. We therefore conclude that these states represent early binding poses, rather than the fully transport-competent conformation.

In these partially occluded states, what determines whether the sugar is to be translocated? The asparagine TM7b residue

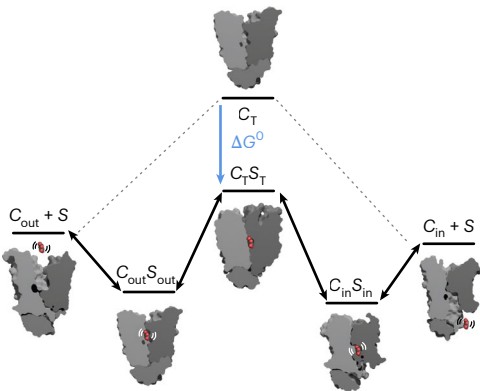

**Fig. 6 | Schematic illustrating a conformational selection-fit model for sugar transport, whereby the binding of a substrate sugar stabilizes the occluded state, which can already be populated spontaneously in the absence of sugar, albeit less frequently.** Structures representing the different states correspond to previously resolved structures: outward-open GLUT3 (PDB 5C65), outward-occluded XylE (PDB 4GC0), occluded *Pf*HT1 (PDB 6RW3), inward-occluded XylE (PDB 4JA3) and inward-open XylE (PDB 4QIQ)[9]. The free-energy diagram was based on thermodynamic measurements of GLUT1 (ref. 48). out, outward; in, inward; $\Delta G$, change in free energy. $C$, conformation; $S$, substrate; T, transition.

corresponding to N311 in *Pf*HT1 and N293 in rat GLUT5 is conserved across all sugar porters, and has been proposed as a key interaction that couples sugar coordination with TM7b gate closure[1]. In both the *Pf*HT1 crystal and cryo-EM structures, 2,5-AHM interacts with N311 Fig. 4c. Q306 initially interacts with the C3–OH group in the inward-occluded state and then shifts to stabiles an interaction with N311 in the occluded state instead. On the other hand, N311 forms more extensive interactions by interacting with both the C3–OH and C4–OH groups in the occluded state. In rat GLUT5 MD simulations, the corresponding asparagine residue formed a stable interaction with D-fructose[25]. Consistently, N311A and N293A variants in TM7b of *Pf*HT1 and rat GLUT5 abolished transport and severely diminished the STD NMR signals (Fig. 5a,e,f, Extended Data Fig. 7a and Supplementary Table 2). Gate closure by TM7b is influenced by residues along its entire length and it was shown previously that the mutation of most TM7b residues to alanine abolished *Pf*HT1 and GLUT5 activity[19,25] (Fig. 5e,f and Extended Data Fig. 7a). The XylE residue L297 is a phenylalanine in GLUT1–4, a tyrosine in GLUT5 and valine in *Pf*HT1. Whereas a V314Y variant in *Pf*HT1 to match GLUT5 severely affected both D-glucose and D-fructose transport and STD NMR signals (Fig. 5a), a Y296F variant in GLUT5 showed similar D-fructose transport activity as wildtype (Fig. 5f and Supplementary Table 2). Although we now have a clearer role of the allosteric coupling by TM7b gate closure, a more exhaustive mutagenesis, and perhaps the investigation of more diffuse properties such as overall protein dynamics[32], will probably be needed to fully shift sugar preferences.

## Conclusion

Sugar transporters have to catalyze the fast flux of sugars at physiologically relevant concentrations that are typically in the millimolar range[2,4,46]. Despite the fact that sugar porters represent one of the largest family of membrane transporters for a wide range of different sugars with varying kinetics[1], the sugar-binding site itself, from parasites to human, has remained remarkably well conserved[1]. Crystal and cryo-EM structures under nonphysiological cryogenic temperatures have enabled the possibility of capturing sugar-bound states of these sugar transporters[8–22], which would otherwise probably be too transient to obtain under physiological conditions. Further, most sugar-bound states also have coligands interacting with either TM7b or TM10b gating helices that have probably helped to stabilize these intermediate

conformations. However, it is important to understand how sugars interact under more physiologically relevant conditions to understand how binding is coupled with TM7b gating and dynamics, which are ultimately deciding which sugars are transported.

By measuring sugar binding to sugar transporters in liposomes by STD NMR at room temperature, we have herein been able to discern that only substrate–sugar pairings that catalyze translocation produce signals. Indeed, MD simulations show that sugars in the outward-occluded state are highly dynamic and although these simulations are not long enough to capture sugar translocation, previous enhanced-sampling MD simulations of GLUT5 indeed showed that the sugar became highly coordinated only upon transition through the occluded state[25]. Consistently, previous [1]H NMR measurements of D-glucose interacting with GLUT1 in native red blood cells concluded that only ~10% fast sugar-binding events led to the formation of a sugar-translocated complex[47], that is, the outward-occluded sugar-bound states are also transient in intact cells. Comparing structures of *Pf*HT1 between inward-occluded and occluded states demonstrate how sugars become better coordinated in transitioning to an occluded formation.

What determines whether these suboptimal partially occluded states catalyze sugar translocation? The difference between a sugar substrate (D-xylose) and the sugar inhibitor (D-glucose) in XylE was not apparent from how the sugars are coordinated in the outward-occluded crystal structures, or from their binding affinities, which are comparable. In fact, mutations in XylE that enable the protein to transport D-glucose did not alter the affinity towards this sugar. Rather, from STD NMR analysis and MD simulations we can conclude that subtle differences enable a substrate sugar (D-xylose) to be highly coordinated for a longer period of time than D-glucose, by better engaging TM7b. The structure of *Pf*HT1 in complex with the D-fructose analog 2,5-AHM confirms the key importance of the TM7b asparagine residue N311 to enable this conformational selection. The mutation of the TM7b residue L297 to phenylalanine to enable XylE to transport glucose is a potent example of the general importance of TM7b for orchestrating whether sugar binding is further able to undergo an induced-fit to catalyze transport.

Kinetic measurements of GLUT1 transport at different temperatures in human red blood cells were used previously to calculate thermodynamic parameters[48]. Although global conformational changes from inward-to-outward states were found to be strongly endothermic, D-glucose binding had only a small change in enthalpy and rather increased entropy[48]. It was proposed that, as D-glucose is highly hydrated it could effectively exchange hydrogen bonds with water for a similar number of hydrogen bonds with the transporter, that is, to be consistent with the small change in enthalpy. It was further concluded that loss of translational freedom associated with D-glucose binding to the transporter would be outweighed by the increased entropy, which arises from the removal of water from both glucose and the substrate-binding pocket[25]. This interpretation is consistent with the decreased size of the sugar-binding pocket between inward-occluded and occluded states of *Pf*HT1. Furthermore, kinetic analysis of GLUT1 in red blood cells and ITC measurements of D-xylose addition to purified XylE have shown that locked states of XylE give rise to only a small enthalpic change ($\Delta H$) of −1 to 3 kcal mol[−1], which would be consistent with weak sugar binding to conformationally select preformed outward-occluded and inward-occluded states[31]. Rather, ITC measurements of the unlocked WT XylE protein has shown that D-xylose binding has a larger entropic change ($\Delta S$) component (25 kcal mol[−1]) with only a small positive $\Delta H$ (2.5 kcal mol[−1])[31]. Thus, sugar translocation is largely entropically driven.

Taken together, we propose that sugar translocation in GLUTs and the larger sugar porter family is a two-step mechanism. First a conformational selection to either an outward- or inward-occluded state, followed by a second induced-fit step into a fully occluded state to catalyze sugar translocation. From this two-step model, we can further conclude that the occluded state should be considered equivalent to the transition state in enzymes, as it has the strongest coordination

for the sugar. The difference is that, rather than catalyzing a substrate into a product, its formation in transporters catalyzes a conformational change to translocate the substrate across the membrane. Notably, some enzymes have also been shown to operate by a two-step mechanism[49]. It follows that the optimal fit of the substrate to the fully occluded (transition) state, rather than to the outward-occluded and inward-occluded states, primarily determines substrate specificity in transporters (Fig. 6). Consistent with our analysis of the sugar porter family, the occluded translocation intermediate in Na$^+$-coupled glutamate transporters was also shown to have a dramatically higher affinity for its (neurotransmitter) substrate[47]. Finally, we suggest that STD NMR should be considered more often in analyzing substrate-coupling mechanisms in transporters, and allosteric substrate-driven gating should be analyzed more closely in general. Similar to enzymes, our work further implies that transition-state inhibitors could be modeled based on the fully occluded state in transporters, opening avenues for new transporter therapeutics.

## Online content

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

## Methods

### Construct design and cloning

The *Pf*HT1 gene construct consisting of residues 1–5 from *Rattus norvegicus* GLUT5 followed by *Pf*HT1 residues 20–504 out of 504 (UniProt accession no. O97467), and *R. norvegicus* GLUT5 full-length gene construct (UniProt accession no. P43427) with a deglycosylation mutation (N50Y) were synthesized and cloned into the GAL1 inducible vector pDDGFP2 containing tobacco etch virus protease (TEV) cleavage site and a C-terminal GFP-His$_8$ tag. The plasmids were transformed into *Saccharomyces cerevisiae*[38,51,52]. The translated sequence after TEV digestion is shown below with the non-*Pf*HT1, *rat* GLUT5 residues on the N terminus and residues remaining of the TEV cleavage site presented in italic. For the *R. norvegicus* GLUT5 construct, the N50Y deglycosylation mutation and C-terminal residues remaining after TEV cleavage are shown in italic. *Pf*HT1: *MEKED*SGFFSTSFKYVLSACIASFIFGYQVSVLNTIKNFIVVEFEWCKGEKDRLNCSNNTIQSSFLLASVFIGAVLGCGFSGYLVQFGRRLSLLIIYNFFFLVSILTSITHHFHTILFARLLSGFGIGLVTVSVPMYISEMTHKDKKGAYGVMHQLFITFGIFVAVMLGLAMGEGPKADSTEPLTSFAKLWWRLMFLFPSVISLIGILALVVFFKEETPYFLFEKGRIEESKNILKKIYETDNVDEPLNAIKEAVEQNESAKKNSLSLLSALKIPSYRYVIILGCLLSGLQQFTGINVLVSNSNELYKEFLDSHLITILSVVMTAVNFLMTFPAIYIVEKLGRKTLLLWGCVGVLVAYLPTAIANEINRNSNFVKILSIVATFVMIISFAVSYGPVLWIYLHEMFPSEIKDSAASLASLVNWVCAIIVVFPSDIIIKKSPSILFIVFSVMSILTFFFIFFFIKETKGGEIGTSPYITMEERQKHMTKSVV*ENLYF*Q. GLUT5: MEKEDQEKTGKLTLVLALATFLAAFGSSFQYGYNVAAVNSPSEFMQQFYYDTYYDRNKENIESFTLTLLWSLTVSMFPFGGFIGSLMVGFLVNNLGRKGALLFNNIFSILPAILMGCSKIAKSFEIIIASRLLVGICAGISSNVVPMYLGELAPKNLRGALGVVPQLFITVGILVAQLFGLRSVLASEEGWPILLGLTGVPAGLQLLLLPFFPESPRYLLIQKKNESAAEKALQTLRGWKDVDMEMEEIRKEDEAEKAAGFISVWKLFRMQSLRWQLISTIVLMTGQQLSGVNAIYYYADQIYLSAGVKSNDVQYVTAGTGAVNVFMTMVTVFVVELWGRRNLLLIGFSTCLTACIVLTVALALQNTISWMPYVSIVCVIVYVIGHAVGPSPIPALFITEIFLQSSRPSAYMIGGSVHWLSNFIVGLIFPFIQVGLGPYSFIIFAIICLLTSIYIFMVVPETKGRTFVEINQIFAKKNKVSDVYPEKEEKELNDLPPATREQ*ENLYF*Q.

All *Pf*HT1 and GLUT5 mutants were generated by overlap PCR, cloned into the pDDGFP$_2$ vector as described for WT. The full-length XylE sequence was amplified by PCR using genomic DNA extracted from *E. coli* MACH1 cells (Thermo Scientific) as a template. The amplified DNA and pWaldo-GFPd vector were digested using restriction enzymes *Nde*I and *Bam*HI and ligated to create a plasmid containing a TEV cleavage site and a C-terminal GFP-His$_8$ tag. The XylE mutants were prepared using a QuikChange lightning site-directed mutagenesis kit (Agilent Technologies).

### Large-scale production and purification

For *Pf*HT1 and GLUT5 production, 12 l of *S. cerevisiae* FGY217 cells were cultivated in −URA medium supplemented with 0.1% (w/v) D-glucose at 30 °C and 150 rpm in 2-l Tunair flasks (Merck). The cultures were induced at an optical density at 600 nm (OD$_{600}$) of 0.6 by the addition of galactose to a final concentration of 2% (w/v). At 22 h after induction, the cells were collected, resuspended in buffer consisting of 50 mM Tris-HCl pH 7.6, 1 mM EDTA, 0.6 M sorbitol, and lysed by mechanical disruption as described previously[53]. XylE was overexpressed using *E. coli* BL21-Gold (DE3) strain (Agilent Technologies) in Luria–Bertani (LB) medium with kanamycin at 50 µg ml$^{-1}$; 6 l of *E. coli* BL21-Gold (DE3) was grown at 37 °C and induced by addition of isopropyl-β-D-thiogalactopyranoside (IPTG) to a final concentration of 0.4 mM at an OD$_{600}$ of 0.5 and harvested after overnight incubation at 25 °C by centrifugation. The pelleted cells were resuspended in 1× PBS buffer before lysis at 25,000 psi using a Constant Systems TS series cell disruptor (Constant Systems). For both *S. cerevisiae* and *E. coli*, the cell debris were spun down by centrifugation at 4 °C and 5,000*g* for 20 min. Membranes were then collected from the supernatant by ultracentrifugation at 4 °C and 195,000*g* for

2 h, homogenized in 20 mM Tris-HCl pH 7.5, 0.3 M sucrose, 0.1 mM CaCl$_2$, flash frozen in liquid nitrogen and stored at −80 °C. The membranes were solubilized for 2 h at 4 °C in equilibration buffer (EB) consisting of 1× PBS, 150 mM NaCl, 10% (v/v) glycerol and 1% (w/v) n-dodecyl-β-D-maltopyranoside (DDM; Glycon). The nonsolubilized material was removed by ultracentrifugation at 195,000*g* for 45 min at 4 °C, and the resulting supernatant was incubated with 15 ml of Ni$^{2+}$-nitrilotriacetate affinity resin (Qiagen) overnight at 4 °C under mild agitation after addition of imidazole to 20 mM. The slurry was poured into a 30-ml Eco-column (Bio-Rad) and washed with 500 ml of EB containing 0.1% (w/v) DDM and 35 mM imidazole. The resin-bound protein was eluted in 30 ml of EB containing 0.02 % (w/v) DDM and 250 mM imidazole. For STD NMR experiments, the eluate was concentrated using 100-kDa molecular-weight cutoff spin concentrators (Amicon Merck-Millipore), flash frozen in liquid nitrogen and stored at −80 °C. For crystallization, the eluate was incubated with equimolar TEV protease at 4 °C overnight to cleave the GFP-His$_8$ tag during dialysis performed against 3 l of dialysis buffer consisting of 20 mM Tris-HCl pH 7.5, 150 mM NaCl and 0.02% (w/v) DDM. The dialyzed and digested sample was loaded onto a 5-ml HisTrap column (GE Healthcare) equilibrated with dialysis buffer and the *Pf*HT1 containing flowthrough was collected and concentrated. The concentrated solution was subjected to a Superose 6 10/300 column (GE Healthcare) pre-equilibrated in a buffer consisting of 20 mM MES pH 6.5, 150 mM NaCl and 0.4% (w/v) β-NG (Anatrace), and the peak fractions were collected and concentrated to 5 mg ml$^{-1}$.

The *Pf*HT1, GLUT5 and XylE mutants were purified as described for WT. The monodispersity of the purified *Pf*HT1, GLUT5 and XylE−GFP fusions was assessed by fluorescence-detection size-exclusion chromatography (FSEC)[53,54] using an Enrich 650 10 × 300 column attached to a Shimadzu HPLC LC-20AD/RF-20A (488 nm$_{excitation}$, 512 nm$_{emission}$) instrument in buffer containing 20 mM Tris-HCl pH 7.5, 150 mM NaCl and 0.03% (w/v) DDM.

### Cysteine crosslinking XylE

Disulfide bond formation was initiated by addition of 1.5 mM CuCl$_2$ to the eluted XylE V35C/E302C mutant. The mixture was agitated mildly at room temperature for 1.5 h on a plate shaker set at 50 rpm. The mixture was dialyzed into buffer containing 25 mM MES pH 6.5, 150 mM NaCl and 0.03% (w/v) DDM for subsequent experiments.

### Crystallization and structure determination of *Pf*HT1

Crystals of *Pf*HT1 in complex with 2,5-AHM were grown at 20 °C using the hanging-drop vapor-diffusion method. Purified *Pf*HT1 protein at 5 mg ml$^{-1}$ was added 2,5-AHM to a final concentration of 50 mM; 1 µl of this solution was mixed 1:1 with reservoir solution consisting of 0.1 M HEPES pH 7.5 and 26% (w/v) polyethylene glycol (PEG) 400. Crystals appeared within 1 week in 26% (w/v) PEG 400 and were collected subsequently and flash frozen in and stored under liquid nitrogen.

X-ray diffraction data of a single *Pf*HT1 crystal was collected at the European Synchrotron Radiation Facility beamline ID23-1, at 100 K and a wavelength of 0.8731 Å. The resulting dataset was indexed, integrated and scaled using XDS[55] before merging using Aimless[56]. Initial phases of *Pf*HT1 were obtained by molecular replacement using Phaser from the Phenix suite[57,58] and the previous structure of *Pf*HT1 in complex with glucose as search model (Protein Data Bank (PDB) 6RW3). The asymmetric unit contained four chains of *Pf*HT1. Structure refinement was carried out using Phenix.refine[59,60] after applying NCS and TLS restraints, interspersed with manual model building in Coot[61]. The R$_{work}$/R$_{free}$ values of the final model were 23.4%/27.6% and the Ramachandran statistics were 96.6% favored, 3.21% allowed and 0.21% outliers. The final model was deposited to the PDB with the accession code 9HKK. Additional data collection and refinement statistics are presented in Table 1. Structural alignments were performed using the align command of PyMol software (http://www.pymol.org/) using Cα coordinates[62,63].

## Reconstitution of membrane protein–GFP fusions into liposomes for STD NMR measurements

Total bovine brain lipid extracts (Sigma-Aldrich) and cholesteryl-hemisuccinate (Sigma-Aldrich) powder were combined by vortexing into a buffer consisting of 25 mM potassium phosphate pH 8.0 and 50 mM NaCl to a final concentration of 30 mg ml$^{-1}$ and 6 mg ml$^{-1}$, respectively. The lipid mixture was subjected to five rounds of freeze–thaw cycles by flash freezing in liquid nitrogen and thawing at room temperature; liposomes were sonicated in 30-s pulses at 40% amplitude, with 10-s cooling intervals between pulses, for a total of six cycles. Lipid mixture was further spun down at 16,000g for 15 min and the supernatant containing small unilamellar vesicles was collected. Purified GFP fusion protein variants (150 µg) were added to 500 µl of unilamellar vesicles, flash frozen and thawed at room temperature. Large, unilamellar proteoliposomes were prepared by extrusion (LiposoFast, Avestin; membrane pore size, 400 nm). The extruded proteoliposomes were collected by ultracentrifugation at 195,000g for 30 min and resuspended in a deuterated buffer consisting of 25 mM potassium phosphate pH 8.0 and 50 mM NaCl. Another round of ultracentrifugation–resuspension steps was applied to ensure a removal of detergents and a complete exchange into the deuterated buffer to obtain well-resolved NMR spectra. The resulting pellet was resuspended in 160 µl of a deuterated buffer consisting of 25 mM potassium phosphate pH 8.0 and 50 mM NaCl. Then, 10 µl of the proteoliposome sample was used for concentration determination and 150 µl for NMR experiments. The same procedure was applied for XylE proteoliposme preparation using *E. coli* polar lipids in a buffer consisting of 25 mM potassium phosphate pH 6.5 and 50 mM NaCl. The concentration of membrane proteins in the proteoliposomes were estimated by GFP fluorescence using a Fluoroskan plate reader (Thermo Fisher, using Skanlt software v.6.0.2) and adjusted to 10 µM for STD NMR experiments.

## STD NMR measurements

The NMR samples were prepared as a mixture of 10 µM membrane protein–GFP fusion reconstituted into proteoliposomes and 500-µM substrates, which were dissolved in buffer containing 25 mM potassium phosphate, 50 mM NaCl in D$_2$O. All NMR experiments were performed at 298 K on Bruker 500 MHz Avance II, 600 MHz Avance III HD or 700 MHz Avance X spectrometers, where the 500 MHz and 700 MHz spectrometers were equipped with 5-mm BBI and BBO cryogenic probes, respectively. NMR spectra were processed using software Topspin v.3.0 (Bruker). On- and off-resonance irradiations were applied at chemical shifts of −0.5 and 60 ppm, respectively. Proteins were saturated using a train of Gaussian-shaped 50-ms-long pulses. The total length of the saturation train was set to 2 s. All NMR spectra were acquired for 18 h with 4,096 scans and a spectral width of 12 ppm.

Protein-ligand binding was quantified by STD amplification factors (STD-AF):

$$STD-AF = (I_0 - I_{sat})/I_0 \times ligand\ excess$$

in which $I_0$ is the ligand peak intensity in an off-resonance $^1$H NMR spectrum, $I_{sat}$ is the intensity in the on-resonance spectrum and $I_0 - I_{sat}$ represents the intensity of the STD NMR spectrum. The ligand excess is given by the ratio of ligand-to-protein concentrations, that is, $[L]_{total}/[P]_{total}$.

## Reconstitution of membrane protein into liposomes for transport assays

The incorporation of rGLUT5 protein variants into proteoliposomes was performed as described recently[25,28]. XylE transport activity was determined by counter flow experiments, and the proteoliposme preparation protocol was therefore modified slightly as followed, before the freeze–thaw cycles, 20 mM D-xylose or 20 mM D-glucose was added to ensure preloading of liposomes with sugar for counterflow experiments. During the reconstitution step, 60 µg of purified XylE-GFP was

added to 500 µl of liposomes. The extruded proteoliposomes were collected by ultracentrifugation at 195,000g for 45 min and resuspended in the same buffer as that in which the lipids were prepared previously in a final volume of 150 µl.

## Uptake of radiolabeled sugars into XylE, *Pf*HT1 and rat GLU5 proteoliposomes

The uptake of radiolabeled sugar by *Pf*HT1 and rGLUT5 was measured as described recently[28]. For counterflow experiment of XylE variants, 5 µl of D-xylose preloaded proteoliposomes were added to 45 µl of assay buffer (20 mM MOPS pH 6.5, 2 mM MgSO$_4$) containing 1 µl of [$^3$H]-D-xylose (10 µM, American Radiolabelled Chemicals) and incubated for 2 min at room temperature before stopping the reaction with 1 ml of assay buffer and filtering through a 0.22-µm filter (Millipore), followed by washing with 6 ml MOPS–MgSO$_4$ buffer. Filters were transferred to scintillation vials, applying 5 ml of scintillation liquid (Ultima Gold, Perkin Elmer) before scintillation detection using a scintillator (TRI-CARB 4810TR 110 V; Perkin Elmer).

For measuring glucose transport activity of XylE WT and mutants, the same procedure as above was performed, but using D-glucose preloaded proteoliposomes and a substrate buffer containing 1 µL of [$^{14}$C]-D-glucose (80 µM, American Radiolabelled Chemicals).

Reported values are from at least triplicates, using empty liposomes as baseline intensity, and WT activity as 100% activity, using D-glucose, D-fructose and D-xylose for 100% activity values for the respective proteins *Pf*HT1, rGLUT5 and XylE.

## MD simulations

All simulations were performed using GROMACS[64] either v.2018.1 (for GLUT3), v.2021.6 or v.2024.1-2. Details of MD simulations of GLUT3 are reported in ref. 19, but in brief the glucose-bound protein was embedded and underwent energy minimization using steepest descent. Equilibration MD was then performed for a total of 375 ps, gradually relaxing positional restraints on protein, POPC lipids and ligands, when relevant. The temperature and pressure were maintained at 303.15 K and 1 bar using the Berendsen thermostat and barostat[65] respectively; the CHARMM36m[63] and the TIP3P[66] forcefields were used.

XylE was embedded in a POPC bilayer using the CHARMM-GUI membrane bilayer builder[67]. The system was solvated in a box of ~100X100X110 nm in presence of 150 mM KCl and the model was built using the CHARMM36m forcefield[68] and the TIP3P water[66].

The system was then equilibrated at 303.15 K and 1 bar using the v-rescale thermostat and c-rescale barostat. Van de Waals interactions were calculated with a cutoff radius of 1.2 nm, whereas electrostatic interactions were calculated using the Particle Mesh Ewald method (PME[69]), with a cutoff of 1.2 nm. Following the standard CHARMM-GUI equilibration pipeline, an energy minimization was followed by 6 steps of equilibration, in which positional restraints on backbone, side chains, dihedral angles and lipids were slowly released. In the final step, side chains, dihedral angles and lipids restraints were deactivated. Five unrestrained 1 µs simulations were then performed for each system, using the v-rescale thermostat[70] and c-rescale barostat[71]. To ensure that the results obtained for GLUT3 could be directly compared to the results more recently obtained in XylE, we repeated the GLUT3 simulations, starting from the glucose-bound structure (4ZW9)[9], and followed the same protocol as for XylE to equilibrate, then five unrestrained 1 µs simulations were performed for each system[19].

Analysis of the gate distances in MD simulations was performed using Python with MD Analysis library. For XylE simulations, the extracellular gate distance was calculated as the distance between the center of mass of residues 28-36 in TM1 and residues 295-301 in TM7. For GLUT3 simulations, the residues used were 28-36 in TM1 and 382-390 in TM7.

Clustering was performed on XylE and GLUT3 simulations to highlight sugar binding poses. Trajectories from different repeats were

concatenated and clustering was performed on the sugar binding residues and sugar coordinates using gmx cluster with the Jarvis Patrick method and a cutoff of 0.08 nm. Figure 3e,f and Extended Data Figs. 2 and 3 show a representative frame for all of the clusters having more than 2 members in XylE and GLUT3 simulations. The number of structures in each of these clusters is reported in Supplementary Tables 3 and 4.

## Cryo-EM sample preparation and data acquisition

A synthetic lipid mixture consisting of 15 mM 1-palmitoyl-2-oleoyl-sn-glycero-3-phosphatidylcholine (POPC; Larodan, cat. no. 37-1618-12), 2.5 mM 1-palmitoyl-2-oleoyl-sn-glycero-3-phosphatidylethanolamine (POPE; Larodan, cat. no. 37-1828-9), and 2.5 mM 1,2-diacyl-sn-glycero-3-phospho-1-D-myo-inositol (POPI; Larodan, cat. no. 37-0132-7) was dissolved in chloroform–methanol (2:1, v/v) and a thin lipid film was prepared using a rotary evaporator. The lipid film was rehydrated in a 20 mM Tris (pH 7.5) and 150 mM NaCl buffer to a final concentration of 20 mM. PfHT1–GFP was combined with MSP1E2, purified as described previously[72] and the lipid mixture POPC:POPE:POPI at a molar ratio of 1:5:100, respectively, and incubated at 4 °C for 30 min. Subsequently, 50 mg of pre-activated SM-2 Bio-Beads (Bio-Rad) was added and incubated at 4 °C overnight to remove detergent. The Bio-Beads were discarded and the supernatant incubated with $Ni^{2+}$-nitrilotriacetate affinity resin (Qiagen) beads to remove unbound free MSPs. The PfHT1 nanodisc sample was eluted in a buffer containing 20 mM Tris-HCl (pH 7.5), 150 mM NaCl and 250 mM imidazole and purified further by size-exclusion chromatography on an Enrich 650 10 × 300 column in a buffer containing 150 mM NaCl and 20 mM Tris-HCl (pH 7.5). The sample was concentrated to ~3 mg ml$^{-1}$, and 50 mM 2,5-AHM was added before grid preparation.

Aliquots of 3-μl PfHT1 nanodiscs were applied to Quantifoil Cu R1.2/0.3 grids and blotted for 5 s at 4 °C under 100% humidity. Grids were plunge-frozen in liquid ethane using a Vitrobot Mark IV (Thermo Fisher Scientific). A total of 17,829 videos were collected on a Titan Krios G3i microscope equipped with a Gatan BioQuantum K3 detector using EPU (Thermo Fisher Scientific). Data collection parameters are summarized in Table 2.

## Cryo-EM data processing

Image processing was performed in CryoSPARC[73] Micrographs were motion-corrected using 'patch motion,' and contrast transfer function (CTF) parameters were estimated using 'patch ctf estimation.' Following CTF estimation, micrographs were denoised before template-based particle picking. Particles near ice or carbon edges were excluded using micrograph junk detector, yielding ~5.7 million particles. After several rounds of two-dimensional classification, 1,262,390 particles were retained for multimodel ab initio reconstruction, resulting in a high-quality class containing 625,315 particles. Subsequent heterogeneous refinements yielded a more homogeneous dataset of 300,678 particles. Final three-dimensional reconstruction using nonuniform and local refinement achieved a gold-standard resolution of 2.42 Å.

## Reporting summary

Further information on research design is available in the Nature Portfolio Reporting Summary linked to this article.

## Data availability

The coordinates and the structure factors for X-ray crystallography structure of PfHT1 have been deposited at PDB 9HKK. The PfHT1 cryo-EM structure has been deposited at PDB 9SDL and EMDB 54787. All data are available in the paper or in Tables 1 and 2. The STD NMR data are available via Zenodo at https://doi.org/10.5281/zenodo.18722732 (ref. 74). The molecular dynamics data is available via Zenodo at https://doi.org/10.5281/zenodo.18347238 (ref. 75). Data and materials can be obtained from the corresponding authors upon request. Source data are provided with this paper.

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

## Acknowledgements

X-ray diffraction data were collected at the European Synchrotron Radiation Facility beamline ID23-1 Grenoble, France. Cryo-EM data were collected the Cryo-EM Swedish National Facility at SciLifeLab Stockholm, Sweden. MD simulations were enabled by the National Academic Infrastructure for Supercomputing in Sweden (NAISS). This work was supported by grants from the Swedish Research Council (2019-02433; 2022-03014) (L.D.; G.W.), The Knut and Alice Wallenberg Foundation (L.D.; D.D.), Novo Nordisk Foundation (D.D.), The Swedish Cancer Foundation (D.D.) and ERC AdG MEMSUGAR (101201464) (D.D.).

## Author contributions

D.D. designed the project and G.W. provided guidance with STD NMR. Expression screening and sample preparation for X-ray crystallography and cryo-EM was carried out by D.-H.A. and A.G. Crystallization and X-ray collection was carried out by D.-H.A. and X-ray refinement and model building by D.-H.A., A.G. and M.C. Cryo-EM data collection and map reconstruction was carried out by A.G. STD NMR was carried out by D.-H.A. with support from A.R., J.M.S. and G.W. Transport assays were carried out by T.R. and A.S. MD simulations were carried out by C.A. with support from M.B. and L.D. The manuscript was written mostly by D.D. All authors discussed the results and commented on the manuscript.

## Funding

## Competing interests

The authors declare no competing interests.

## Additional information

**Extended data** is available for this paper at https://doi.org/10.1038/s41594-026-01784-w.

**Correspondence and requests for materials** should be addressed to Göran Widmalm or David Drew.

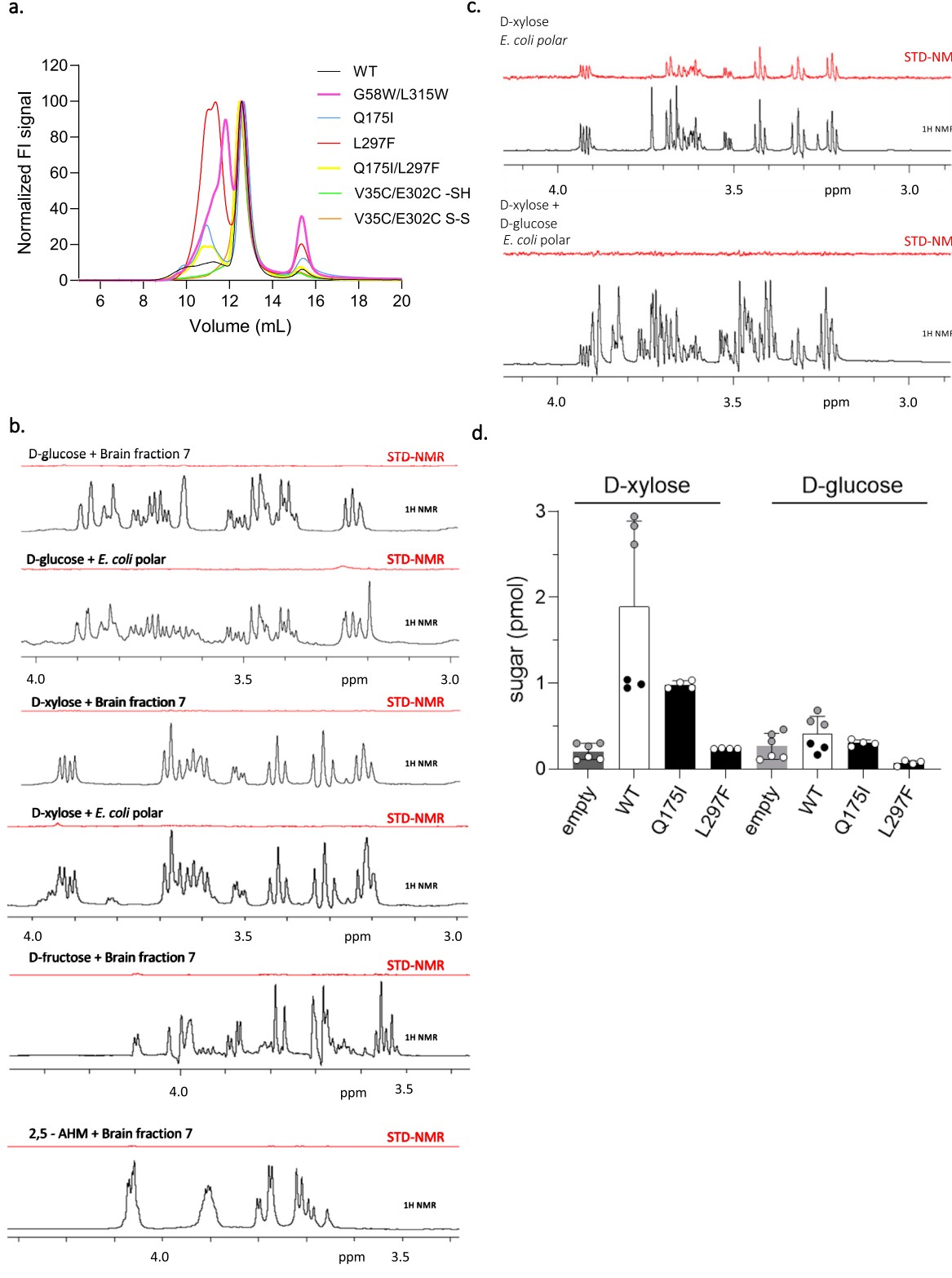

**Extended Data Fig. 1 | FSEC traces of XylE constructs, STD NMR control and competition experiments, and XylE sugar binding or gating helix mutant transport assay. a**. FSEC traces of DDM purified XylE-GFP constructs. **b**. STD NMR data from protein-free liposomes of different lipid composition in the presence of sugars. **c**. Competition between D-xylose and D-glucose measured by STD NMR using WT XylE. (top) [1]H and STD spectrum with only D-xylose. (bottom) as top but after addition of D-Glucose to the same sample, showing the loss of STD signal after addition of the inhibitory sugar. **d**. Uptake of [3]H-D-xylose or [14]C-D-glucose by XylE WT or the mutants Q175I and L297F in proteoliposomes. Error bars indicate mean ± s.e.m. of n 4 or 6 independent experiments.

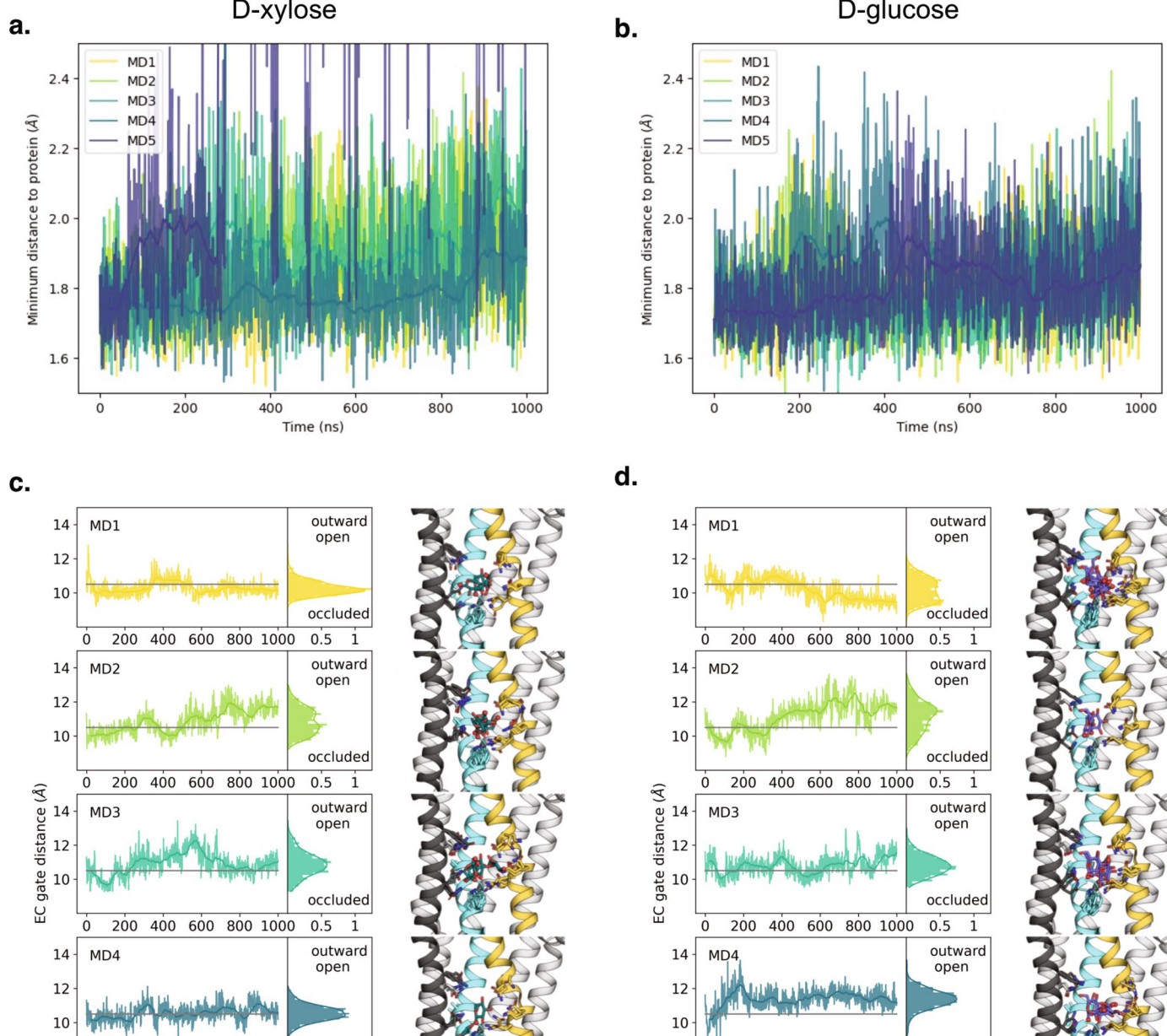

**Extended Data Fig. 2 | Extracellular gate distances overtime and sugar positions for five repeats. a.** Distance of D-xylose and **b.** D-glucose from the center of mass of the protein in five repeats, indicating if the sugar is leaving the protein during the simulation time; **c.** Extracellular gate distances overtime per repeat and structures belonging to the cluster corresponding to the starting pose, that is the one resolved in the starting structure (in blue) for five repeats of D-xylose-bound and **d.** D-glucose-bound XylE.; the gray line represents the gate distance at which the extracellular gate is considered as 'occluded'.

**a.**

GLUT3

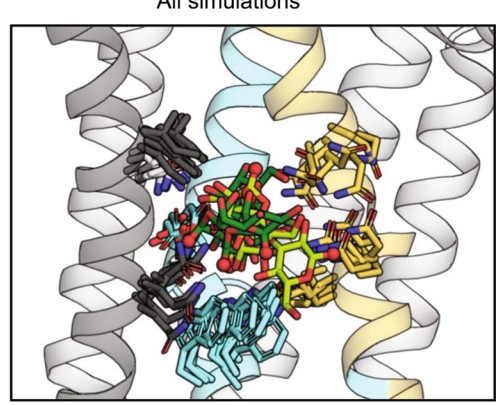

**b.**

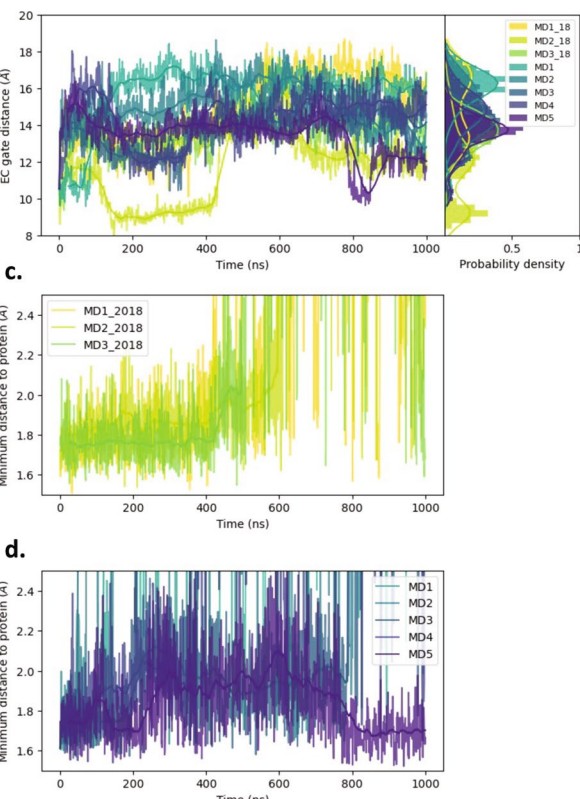

**c.**

**d.**

**Extended Data Fig. 3 | GLUT3 MD simulations. a**. Position of D-Glucose (green sticks) and coordinating residues (shown as sticks) in the frequently populated clusters in three 1 μs-long simulations using gromacs 2018 (light green sticks) and gromacs 2024 (dark green sticks) bound structure of GLUT3 (PDB: 4ZW9) (protein shown as cartoon). TM1, TM2 and TM4 were omitted for clarity, the O1 atom is highlighted by a red sphere for reference; **b**. Extracellular gate distances overtime for eight repeats; **c**. D-glucose distance from the center of mass of the protein in three repeats using gromacs 2018 and **d**. gromacs 2024.

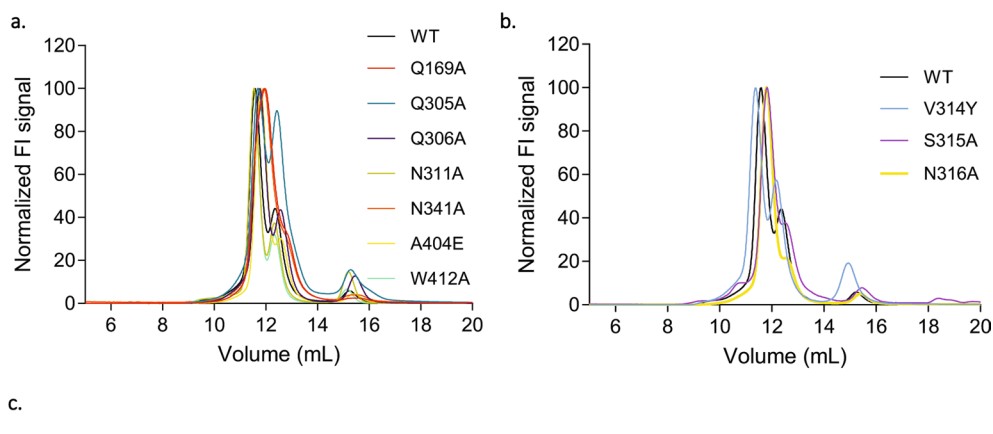

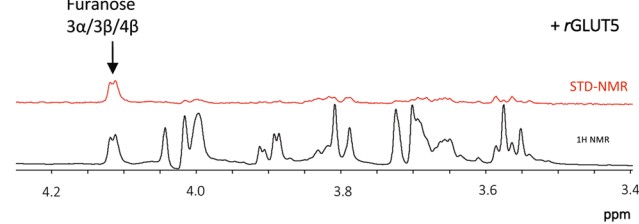

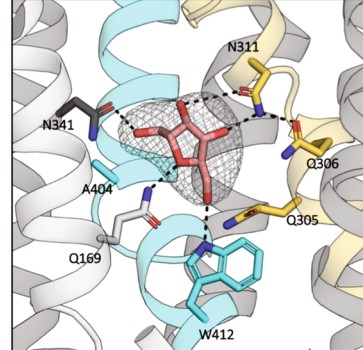
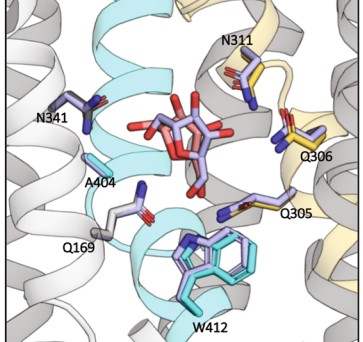

**Extended Data Fig. 4 | FSEC traces of *Pf*HT1 constructs, rat GLUT5 interaction with D-fructose measured by STD NMR and crystal structure of *Pf*HT1 in complex with 2,5-AHM.** **a**. FSEC traces of DDM purified sugar-binding site mutants of *Pf*HT1-GFP fusions. **b**. FSEC traces of DDM purified TM7b mutants of *Pf*HT1-GFP fusions. **c**. The structure of D-fructofuranose and corresponding STD NMR (red) and ¹H NMR (black) spectra in the presence of rat GLUT5-GFP reconstituted into liposomes, with the STD signal producing protons shown on red background. **d**. (left) binding site of the *Pf*HT1 X-ray complex with 2,5-AHM (pink sticks), and hydrogen bonding residues (sticks), and hydrogen bond interactions shown as dashed lines overlayed with a Podler omit electron density map (contoured at 5σ). (right). Comparison of the two slightly different 2,5-AHM poses observed in the X-ray structure, pairwise represented by chains A and C *vs*. chains B and D, within the asymmetric unit containing four copies of *Pf*H1, with protein shown as in Fig. 1b .

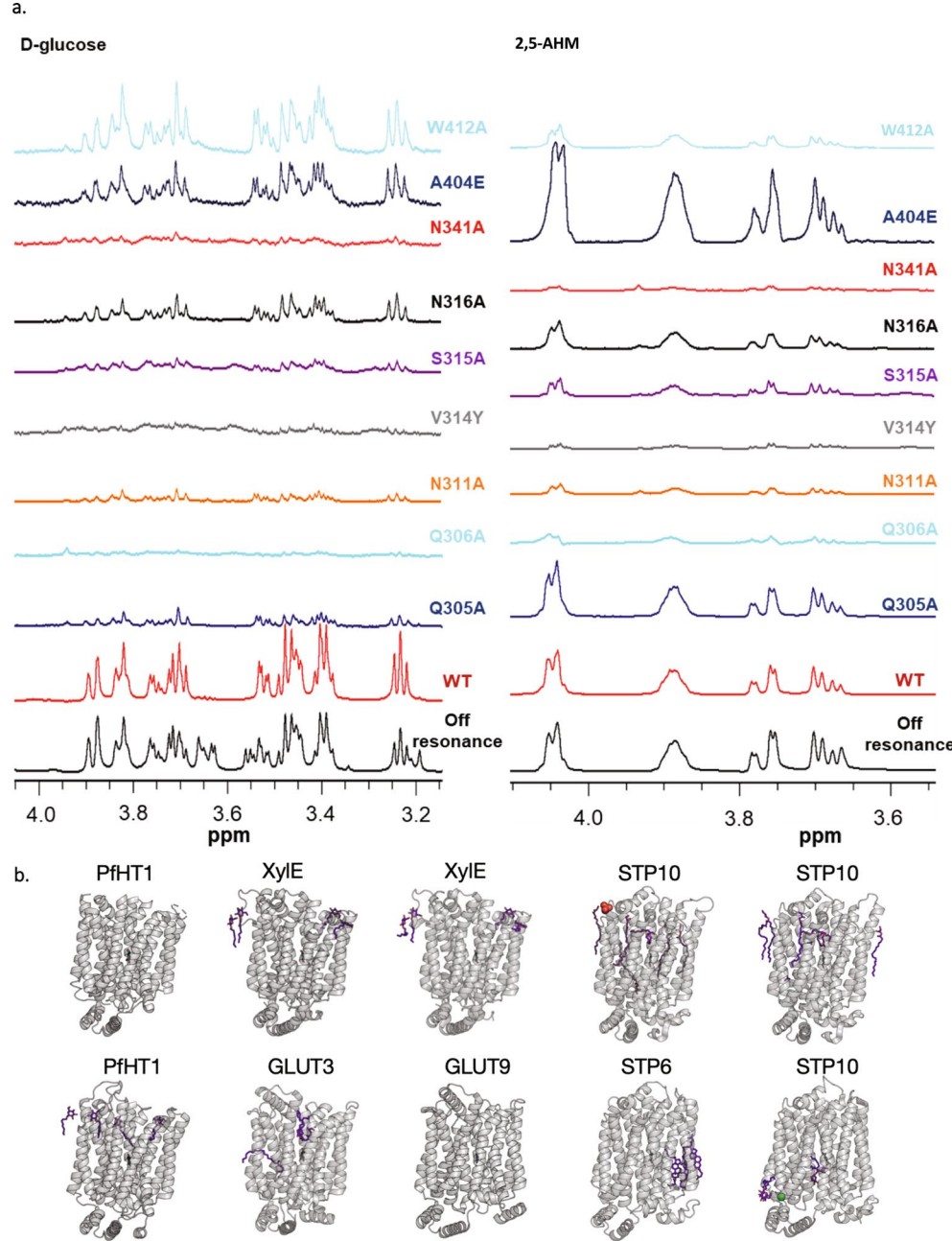

**Extended Data Fig. 5 | STD NMR of *Pf*HT1 constructs and sugar transporter structures determined in the presence of substrates or a substrate-like molecules. a**. (left) STD (colored) and ¹H spectra (black) of D-glucose in the presence of WT (red) or sugar-binding site mutants of *Pf*HT1 (colored individually). (right) As in the left panel but for 2,5-AHM. **b**. Sugar transporters structures resolved in complex with a substrate or substrate-like molecule are presented, from the top left *Pf*HT1 (PDB: 6RW3), XylE (PDB: 4GBY), XylE (PDB: 4GBZ), STP10 (PDB: 6H7D), STP10 (PDB: 7AAQ), from the bottom left *Pf*HT1 (PDB: 6M20), GLUT3 (PDB: 7SPT), GLUT9 (PDB: 8Y65), STP6 (PDB: 9G11), STP10 (PDB: 7AAR), with protein shown as cartoons in gray, the ligand as sticks in teal and the lipids/detergents present to stabilize the structure as purple sticks.

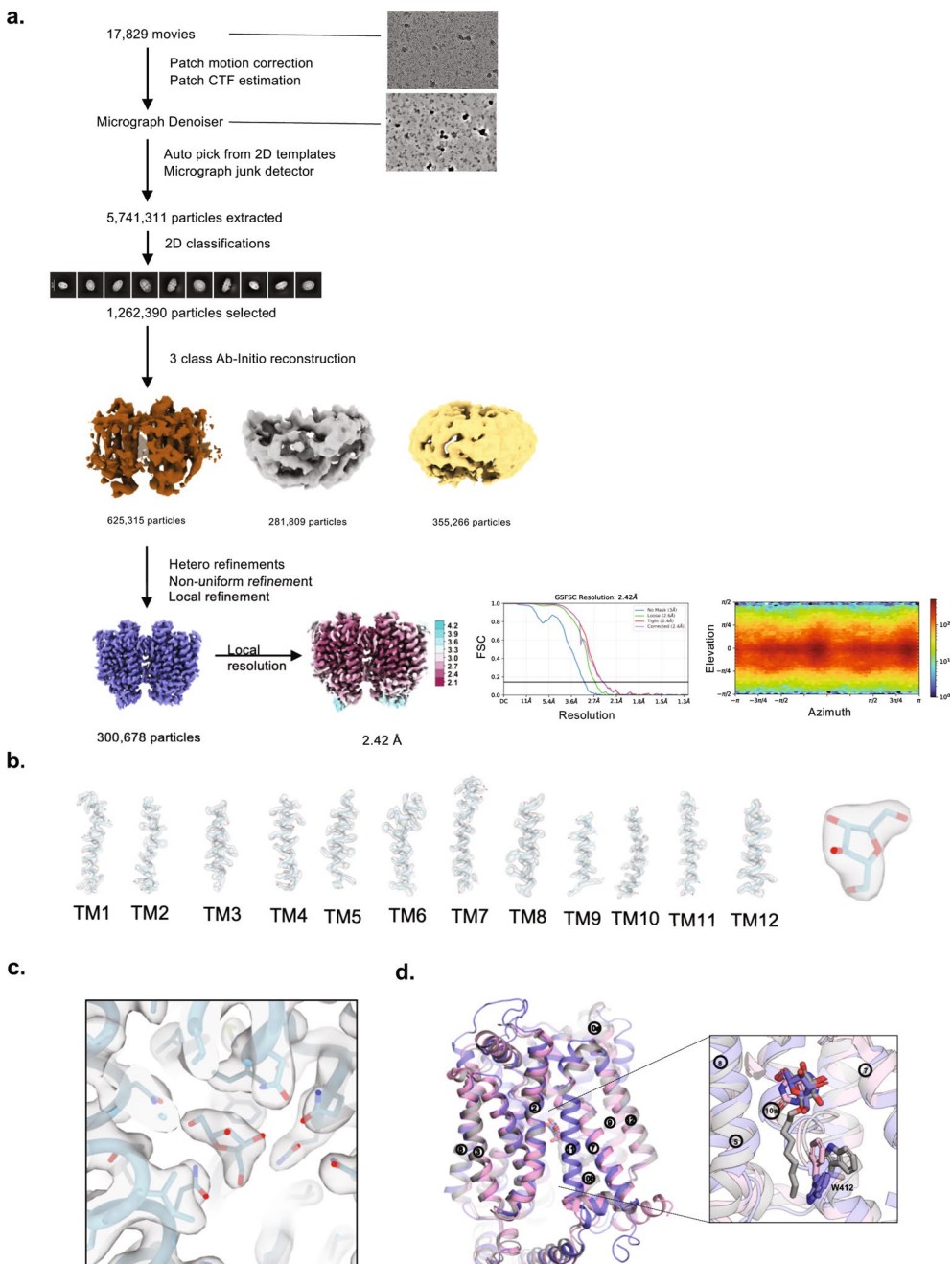

**Extended Data Fig. 6 | Cryo-EM processing workflow of *Pf*HT1 with 2,5-AHM.**
**a**. Data was processed using CryoSPARC[73]. Movie frames were aligned using the "Patch motion correction" and CTF was estimated by "patch CTF" algorithms. Micrographs were denoised before particle picking and picks near ice and aggregation was removed using micrograph junk detector. The dataset was initially cleaned using multiple rounds of 2D classifications, initial maps were generated using multiclass *ab initio* reconstruction and further homogenous particle set was obtained using heterogenous refinement. Final *Pf*HT1 cryo-EM map with 2,5-AHM was reconstructed from around 300,000 particles after non-uniform and local refinement, with an overall resolution of 2.42 Å resolution according to the FSC at 0.143. **b**. Cryo-EM maps for all transmembrane helices of this *Pf*HT1 structure (transparent grey surface). Modelled residues and 2,5-AHM are shown as cyan sticks, and protein as cartoon also in cyan. **c**. Cryo-EM densities of 2,5-AHM and neighbouring residues at a map threshold of 0.075, colored as in b. All densities and volumes were rendered using ChimeraX[76]. **d**. (left) Structural superposition of inward-occluded 2,5-AHM bound *Pf*HT1 structure (blue) with inward-open urate bound GLUT9 (pink; PDB:8Y65) and β-NG bound GLUT1 (gray; PDB: 4PYP) structures. (right) zoomed in view of the binding pocket. Protein shown as cartoon with selected residues and ligands as sticks, all colored as per structure.

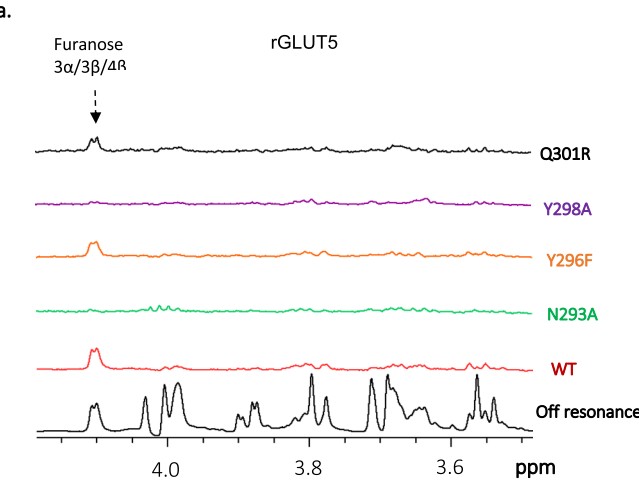

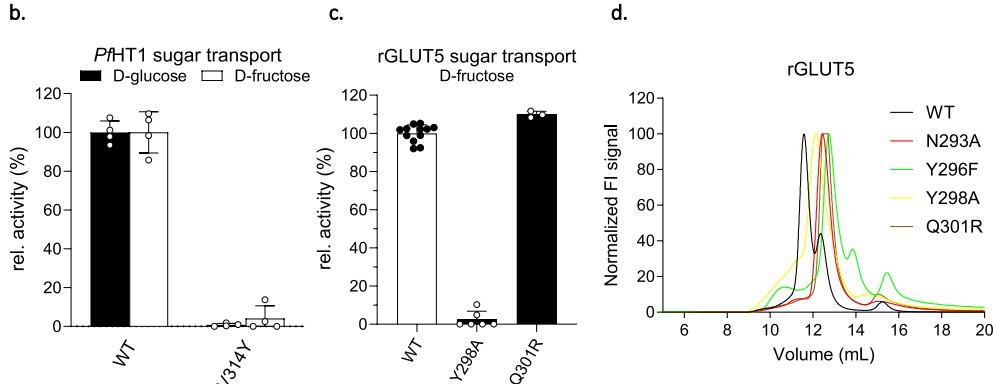

**Extended Data Fig. 7 | STD NMR and functional analysis of rat GLUT5 and *Pf*HT1 TM7b gating residues. a**. STD NMR (colored) and ¹H spectra (black) of D-fructose in the presence of WT (red) or TM7b mutants of rat GLUT5. **b**. Normalized uptake of ¹⁴C-D-glucose or ¹⁴C-D-fructose by WT *Pf*HT1 and the mutant V314Y in proteoliposomes prepared using brain-fraction-seven

lipids. Error bars indicate mean ± s.e.m. of n ≥ 3 independent experiments. **c**. Normalized uptake of ¹⁴C-D-fructose by rat GLUT5 and the TM7b gating mutant Y297A. Error bars indicate mean ± s.e.m. of n ≥ 3 independent experiments. **d**. FSEC traces of DDM purified rat GLUT5 WT and variants.

# Reporting Summary

## Statistics

For all statistical analyses, confirm that the following items are present in the figure legend, table legend, main text, or Methods section.

| n/a | Confirmed | |
|---|---|---|
| ☐ | ☒ | The exact sample size (*n*) for each experimental group/condition, given as a discrete number and unit of measurement |
| ☐ | ☒ | A statement on whether measurements were taken from distinct samples or whether the same sample was measured repeatedly |
| ☒ | ☐ | The statistical test(s) used AND whether they are one- or two-sided *Only common tests should be described solely by name; describe more complex techniques in the Methods section.* |
| ☒ | ☐ | A description of all covariates tested |
| ☒ | ☐ | A description of any assumptions or corrections, such as tests of normality and adjustment for multiple comparisons |
| ☐ | ☒ | A full description of the statistical parameters including central tendency (e.g. means) or other basic estimates (e.g. regression coefficient) AND variation (e.g. standard deviation) or associated estimates of uncertainty (e.g. confidence intervals) |
| ☒ | ☐ | For null hypothesis testing, the test statistic (e.g. *F*, *t*, *r*) with confidence intervals, effect sizes, degrees of freedom and *P* value noted *Give P values as exact values whenever suitable.* |
| ☒ | ☐ | For Bayesian analysis, information on the choice of priors and Markov chain Monte Carlo settings |
| ☒ | ☐ | For hierarchical and complex designs, identification of the appropriate level for tests and full reporting of outcomes |
| ☒ | ☐ | Estimates of effect sizes (e.g. Cohen's *d*, Pearson's *r*), indicating how they were calculated |

*Our web collection on statistics for biologists contains articles on many of the points above.*

## Software and code

Policy information about availability of computer code

| Data collection | MXCube3<br>Bruker Topspin v3.0 |
|---|---|
| Data analysis | Prism 7.04 - for data plotting and analysis<br>Phenix v1.20.1-4487 - Structural refinement software suite<br>PyMol v3.0.1 - Molecular graphics software<br>Coot v0.9.8.92 EL- Structural model building<br>Bruker Topspin v3.0<br>CCP4 7.0 |

For manuscripts utilizing custom algorithms or software that are central to the research but not yet described in published literature, software must be made available to editors and reviewers. We strongly encourage code deposition in a community repository (e.g. GitHub). See the Nature Portfolio guidelines for submitting code & software for further information.

## Data

Policy information about availability of data

All manuscripts must include a data availability statement. This statement should provide the following information, where applicable:

- Accession codes, unique identifiers, or web links for publicly available datasets
- A description of any restrictions on data availability
- For clinical datasets or third party data, please ensure that the statement adheres to our policy

> The coordinates and the structure factors for PfHT1 in complex with 2,5-DHM have been deposited in the Protein Data Bank with accession: 9HKK

## Research involving human participants, their data, or biological material

Policy information about studies with human participants or human data. See also policy information about sex, gender (identity/presentation), and sexual orientation and race, ethnicity and racism.

| | |
|---|---|
| Reporting on sex and gender | n/a |
| Reporting on race, ethnicity, or other socially relevant groupings | n/a |
| Population characteristics | n/a |
| Recruitment | n/a |
| Ethics oversight | n/a |

Note that full information on the approval of the study protocol must also be provided in the manuscript.

# Field-specific reporting

Please select the one below that is the best fit for your research. If you are not sure, read the appropriate sections before making your selection.

☒ Life sciences ☐ Behavioural & social sciences ☐ Ecological, evolutionary & environmental sciences

For a reference copy of the document with all sections, see nature.com/documents/nr-reporting-summary-flat.pdf

# Life sciences study design

All studies must disclose on these points even when the disclosure is negative.

| | |
|---|---|
| Sample size | Biochemical assays and were typically performed at least in triplicate (n =3) to ascertain accurate values for data shown. Statistical methods were not used to determine sample size, but the number of repeats were chosen based on standard practice in the transporter biochemistry community, and were sufficient to calculate standard deviations or standard error. The STD NMR technique was used to screen different combinations of sugars and proteins as well as mutants thereof and as such 4096 scans were acquired per NMR experiment, which was performed once for each mixture of protein and ligand. |
| Data exclusions | No data was excluded. |
| Replication | All biochemical assays were repeated at least 3 times and the results were reproduced each time. |
| Randomization | Blinding was not relevant to our study for biochemical, biophysical and structural analysis |
| Blinding | No blinding was carried out for biochemical and structural analysis as this is not applicable as the analysis does not include subjects. |

# Reporting for specific materials, systems and methods

We require information from authors about some types of materials, experimental systems and methods used in many studies. Here, indicate whether each material, system or method listed is relevant to your study. If you are not sure if a list item applies to your research, read the appropriate section before selecting a response.

## Materials & experimental systems

| n/a | Involved in the study |
|-----|----------------------|
| ☒ ☐ | Antibodies |
| ☒ ☐ | Eukaryotic cell lines |
| ☒ ☐ | Palaeontology and archaeology |
| ☒ ☐ | Animals and other organisms |
| ☒ ☐ | Clinical data |
| ☒ ☐ | Dual use research of concern |
| ☒ ☐ | Plants |

## Methods

| n/a | Involved in the study |
|-----|----------------------|
| ☒ ☐ | ChIP-seq |
| ☒ ☐ | Flow cytometry |
| ☒ ☐ | MRI-based neuroimaging |

## Plants

| | |
|---|---|
| Seed stocks | N/A |
| Novel plant genotypes | N/A |
| Authentication | N/A |

