## [Peer Review File · Nature Structural & Molecular Biology]

A Two-step Mechanism for Sugar Translocation

Corresponding Author: Professor David Drew

Version 0:

Decision Letter:

14th Mar 2025

Dear Dr. Drew,

Thank you again for submitting your manuscript "A Conformational-Selection-Fit model for Sugar Transport". I apologize for the delay in responding, which resulted from the difficulty in obtaining suitable referee reports. Nevertheless, we now have comments (below) from the 3 reviewers who evaluated your paper. In light of those reports, we remain interested in your study and would like to see your response to the comments of the referees, in the form of a revised manuscript.

You will see that while reviewers appreciate the results, they raise several concerns which will need to be addressed in a revision. Specifically, reviewer #1 asks to address the discrepancies between ITC and NMR data. This reviewer suggests the use of further controls in NMR experiments, as well as collecting data at higher concentrations. We agree that strengthening the NMR data and its interpretations will be important. Reviewer #2 also requests further controls with additional sugars, but also expresses concerns regarding the resolution of crystallography data. We agree that if possible, the increased resolution will strengthen the conclusions. Please also discuss the newly proposed model in more details, as per reviewer #2 suggestion. Finally, we also agree with reviewer #3 comments regarding the kinetic data. This reviewer echoes some concerns of referee #1 regarding STD NMR ability to detect transport, not binding, and we ask that you address those. Please address the reviews in full to the best of your ability, and discuss any potential caveats of the approaches as needed.

Please be sure to address/respond to all concerns of the referees in full in a point-by-point response and highlight all changes in the revised manuscript text file. If you have comments that are intended for editors only, please include those in a separate cover letter.

We expect to see your revised manuscript within 3-6 months. If you cannot send it within this time, please contact us to discuss an extension; we would still consider your revision, provided that no similar work has been accepted for publication at NSMB or published elsewhere.

Reporting Summary:

- that unprocessed scans are clearly labelled and match the gels and western blots presented in figures.
- that control panels for gels and western blots are appropriately described as loading on sample processing controls

-- all images in the paper are checked for duplication of panels and for splicing of gel lanes.

EXTENDED DATA FIGURES

Please note that all key data shown in the main figures as cropped gels or blots should be presented in uncropped form, with molecular weight markers. These data can be aggregated into a single supplementary figure item. While these data can be displayed in a relatively informal style, they must refer back to the relevant figures. These data should be submitted with the final revision, as source data, prior to acceptance, but you may want to start putting it together at this point.

REQUIREMENTS FOR REPORTING MD SIMULATIONS:

Please complete the MD simulations checklist provided with this email and provide any additional data on resubmission or deposit all code associated with the paper in a persistent repository where they can be freely and enduringly accessed. Please include a statement under the heading 'Code Availability', indicating whether and how the code can be accessed, including any restrictions to access. If the code can only be shared on request, please explain why in your Code Availability Statement and in your response here.

Please refer to the following editorial: <https://www.nature.com/articles/s42003-023-04653-0>

REQUIREMENTS for reporting protein NMR:

Please provide a full description of the NMR experiments in the Methods. This should include

- Details of the spectrometer(s), additional hardware (such as a cryoprobe), and software (including the version number) used to acquire the data. For solid state NMR experiments, probes, the size of the rotor, the spinning frequency and the temperature of the experiments should be also indicated.
- Details of all pulse sequences used and their corresponding references. Please consider providing the spectral width, acquisition times and number of scans used along with any other parameters that have been optimised during data collection. When spectrometer referencing has been performed please consider providing detail.
- Please provide detail of the program(s) used for data analysis and spectra processing, including version numbers.
- Please ensure all sample concentrations and buffer conditions are reported as well as the temperatures used for recording these data, preferably in kelvin.
- Please report mixing and delay times for all NOESY based experiments reported in your manuscript.
- For solid state NMR experiments, please indicate the contact times for CP steps and the channels frequency for the optimization of the Hartmann-Hahn condition. Please consider providing delay times and mixing times used for CP transfer along with any other parameters that have been optimised during data collection.

ADDITIONAL REQUIREMENTS FOR PROTEIN BACKBONE ASSIGNMENTS AND RELAXATION EXPERIMENTS:

- If reporting a novel protein backbone assignment, please make these data available in the BMRB [<https://bmr.io/>], report the percentage sequence coverage in the main text and provide an assigned spectrum as a full-page Supplementary Figure.

Please consider providing an additional Supplementary Figure highlighting assignment coverage on the protein sequence. Please ensure experimental details including the double/triple resonance spectra used are provided in the Methods.

- If chemical shifts were transferred, please reference the paper and/or BMRB accession number from which these derive. Please ensure a description is provided in the Methods detailing how these shifts were transferred and considerations taken such as buffer matching or additional experiments to minimise errors.

- If a titration is reported, please report each point of the titration series or the steps between each point with the start and end points detailed. Please provide the method used to calculate chemical shift perturbations and provide subsequent data in the Source Data file.

- If reporting relaxation experiments such as NOEs and T1/T2, please ensure that relaxation and interscan delays are reported. Please see <https://www.nature.com/articles/s41467-023-38315-w#Sec8> as an example of how to report these data.

- If any combination of these experiments are reported, please deposit these data to the BMRB [<https://bmr.io/>], and reference the accession number in the data availability statement.

Data availability: this journal strongly supports public availability of data. All data used in accepted papers should be available via a public data repository, or alternatively, as Supplementary Information. If data can only be shared on request, please explain why in your Data Availability Statement, and also in the correspondence with your editor. Please note that for some data types, deposition in a public repository is mandatory - more information on our data deposition policies and available repositories can be found below:

<https://www.nature.com/nature-research/editorial-policies/reporting-standards#availability-of-data>

Link Redacted

Sincerely,

Katarzyna Ciazynska, PhD
(she/her)
Senior Editor
Nature Structural & Molecular Biology
<https://orcid.org/0000-0002-9899-2428>

Referee expertise:

Referee #1: STD NMR

Referee #2: GLUT transporters, structural biology

Referee #3: transporters, computational biology

Reviewers' Comments:

Reviewer #1 (Remarks to the Author):

The authors report on studies about the question on how GLUT transporters achieve sugar specificity, and propose that using STD NMR spectroscopy it is possible to discern between substrates and inhibitors as only transported sugars produce STD NMR signals. An elegant use of mutant seems to support that STD NMR signals is reporting only on transported substrates. Additionally, the results are discussed in the light of MD simulations and X-ray structure for the binding to PfHT1 transporter and rat GLUT5, supporting that TM7b residues control the formation of occluded state. The work present a high degree of originality in the claim that STD NMR spectroscopy can be a novel tool to characterise transport in sugar porters. The quality of the data and the methodology is high, although the interpretation of the results can be a matter of discussion, as I elaborate below.

In particular, I comment below about all the concerns I find for this work to be published in Nature Structural & Molecular Biology, whereby without a proper response to them I would see difficult to get this work published:

It seems that the claim that STD NMR is a tool to report on sugar transport is something that cannot be expressed as a generalization as it may well be simply a particularity of the study systems. In that sense, I can identify a few drawbacks related to the applicability of the technique for detecting transport that the authors should consider:

- How is it possible that, from ITC, binding is measured for the substrate and the inhibitor to be both in the micromolar range, but the binding of the inhibitor is not detected by STD NMR. That affinity is appropriate for binding detection by STD NMR. Is ITC showing binding but STD NMR not? This is difficult to reconcile for the given affinities.
- If the application of STD NMR to distinguish transport or inhibition is simply based on the interpretation of a "significant" reduction of the STD responses in the case of the inhibitor, the methodology seems extremely hard to be properly applied or, better to say, the results properly interpreted. In that sense: what is a "significant" reduction? STD NMR is affected by different things, related not only to the kinetics of the thermodynamics of the process of binding, but also related to differences in sample preparation (e.g. it would be nice to repeat the study of xylose binding with different proteoliposome prepared samples, to check the reproducibility of the STD level of intensities).
- Unless the values of the STD NMR intensities are given, and a "factor" quantifying the differences in level of intensities between substrates and inhibitors, it looks as if the interpretation is in the eye of the beholder (...is it an observed reduction in STD NMR significant as to claim that there is no transport?). Again, it's worth mention that comparing different samples is tricky as during the preparation process the final sample conditions might differ leading to different STD responses. Could competition experiments on a single proteoliposome sample determine if one ligand is observed and not the other? For example, a sample of XylE containing xylose could be put in competition with increasing amounts of glucose, and viceversa.
- I cannot see why binding to these receptors, even in the absence of transport, can never be appropriate for STD NMR observation (in fact that is what I would expect for the studied system, taking consideration of the affinities measured by ITC).
- The concentrations of the ligands are low for STD NMR standards. To be on the safe side about the claim that inhibitors that do not produce transport do not give STD NMR signals, I would recommend running experiments with at least 1 or 2 mM of substrate or inhibitor. In low concentrated ligand samples, STDs might be obscured by the spectral noise, something that would be much improved under larger ligand concentrations. That is the main reason why I propose that the authors should provide a table with the measured STD intensities.
- Figure 1f: what do the authors mean here by "normalized STD effect"? How can it be 0-100 scale for xylose in liposomes from E.coli and only close to 40% in brain lipids fraction? I do not understand how the scales can be transferable between so different sample conditions. Even if the authors want to show that STD responses are much higher in liposomes than in brain lipids, that cannot be correlated directly to transport, as both samples are so different in nature that one would expect, in fact, to get very different STD NMR responses (e.g. the saturation of the proteins can be different using the same irradiation frequency, due to differences in particle sizes, compacity, etc...). This concern is also transferable to the right column bar graphs in Figure 2.
- It is necessary to run all the STD NMR experiments on "blank samples" as controls (samples of liposomes devoid of proteins), under increasing concentrations of ligands (in particular after the proposal from this reviewer of increase the ligand concentrations in the experiments). This is to discard the possibility of having different proportions of non-specific binding to the liposomes by Glc or Xyl that might be impacting the STD results.

- Comparing the STD NMR spectra of glucose binding in figures 1e and 2c, one might get the wrong conclusion that the interpretation of binding in the figure 2c and not binding in figure 1e is just a matter of zooming the Y-axis of the spectra. This is again the reason why it is fundamental that the authors provide tables with the experimental intensities of the STD NMR spectra, to evaluate the ability for discrimination of the proposed approach between substrates and inhibitors.

- Figures 2a and 2b: why do not the authors provide also the studies with Glc, to keep consistency with all the work presented?

- MD simulations were carried out on outward-occluded states of the receptor. I propose also running the MD simulations with the outward-open state of the receptor, as it may well be responsible also for STD NMR responses and corresponding differences in the observed intensities (e.g. Glc being more unstable in that state?).

- How do the authors confirm that all the receptors in the proteoliposomes are well oriented for the transport?

Lines 202 – 205: “Rather than by difference in substrate affinities, we conclude that low intensity STD NMR signals are a result of glucose failing to stabilize an occluded conformation in XylE, which would allow for a long enough interaction time with coordinating residues for sugar protons to achieve sufficient saturation transfer.” And also Lines 322 – 23: “Rather, from STD NMR analysis and MD simulations we can conclude that subtle differences enable a substrate sugar (D-xylose) to be coordinated for a longer period of time than glucose, by which D-xylose can better engage with TM7b”.

This reasoning poses a fundamental thermodynamics/kinetics question: as I understand, in fact, a longer interaction time is associated to a higher affinity. If, for a given ligand to a receptor the STDs are stronger because of a longer “contact time”, the affinity for that ligand should accordingly be stronger, however, both, substrate and inhibitor show rather similar affinities. There is something fundamental related to the claim that I cannot understand.

Line. 209:

Why do the authors use the word “coupling” between glucose and fructose?

Line 215:

Why “Suppl. Fig. 2a” is mentioned here?

Line 217: “more robustly” is not a proper wording for describing the presence of strong STD signals. I would recommend “more closely”

Lines 218 – 219: “The STD NMR analysis is further consistent with the hydrogen bonding distance between the protons in D-glucose and the sugar coordinating side-chains.”

This is interesting, as it seems that they are referring to the orientation in the occluded state, isn't it? If that is the case, how can STD NMR results be reporting on transport, but the epitope reported is related to the binding site in the occluded state? In line with this concern, I would recommend the authors to try to measure binding epitopes for inhibitor, even if the intensities are very low, as it might well be that the resulting epitope will equally be reporting on the binding as observed in the 3D structures.

Lines 264 -266: “Alanine substitutions of the sugar coordinating residues Q305, Q306, N311, N341 all showed low intensity STD NMR signals Fig. 5a consistent with transport assays that have demonstrated these variants abolished D-glucose transport”

Again, this is a clear case were the claim that low intensity indicates no transport doesn't seem to be so strong taking into consideration the evidences in Figure 5a (again, what is the meaning of “normalized STD effects”? (how is scale transferable between samples?)). What is more, in Suppl. Fig. 3, the alanine substituted mutants seems to show clear STD NMR responses. Given these evidences, the claim that STD NMR reports on transport doesn't seem to be clearly supported.

Reviewer #2 (Remarks to the Author):

The authors focused on studying sugar transporters for many years and have published several important papers on the glucose transport mechanism. This work explores the conformational selection-fit model for sugar transport to elucidate how glucose transporters (GLUTs) achieve sugar specificity. The authors employ saturation transfer difference (STD) NMR spectroscopy to analyze the differences between sugar substrates and inhibitors, revealing the molecular basis for the transport mechanism. The study highlights the importance of specific residues in controlling the formation of the occluded

state, concluding that the optimization of substrate interactions with this state primarily determines substrate specificity in transporters. Before accepting, the authors should address the following concerns :

1. The author hypothesized that "only transported sugars were giving rise to STD NMR signals." Some sugars that cannot be bound/transported by XyleE should be considered for measurement by STD NMR as additional controls.
2. The transport assays in Fig2a and Fig2b contain two almost identical WT groups (control). Please recheck it. The authors should provide data from the two individual experiments in the two graphs.
3. The author makes a detailed explanation in the conclusion. However, the difference between conformational selective-fit model and previously accepted models is less clear.
4. The resolution of the crystal structure is slightly low. The local density of small molecules is also not very good. I'm worried that the small molecules are facing the wrong orientation. Can authors collect more diffraction data sets and solve the structure with higher resolution?

Reviewer #3 (Remarks to the Author):

The manuscript by Ahn, et al. describes primarily STD NMR results to conclude that sugar specificity in glucose transporters (GLUTs) is mainly controlled by the ability of the substrate to form an occluded state of the transporter, not by their binding affinity to the protein. They use several proteins for their measurements, including XyleE, a bacterial homolog of GLUTs, which is actually the main system studied here. MD simulations are also used to make sense of the data. MD results show that glucose in the binding site of XyleE locks the protein whereas xylose produces the occluded state, though this observation is not consistent, and in some cases the sugar substrate leaves the binding site. The study is complemented by structural determination of another bacterial homolog. In general, the study reports interesting results, and I lean towards recommending it for publication, but the presentation is rather poor and requires major revision. Below are some specific comments about the manuscript.

- As I indicated, the text is a mess and needs critical reading and revision for typos and grammatical errors. Some of these have been mentioned below in my minor comments.
- At points not enough information is provided to get a clear idea about what has been done, particularly in the methods section. More details can be offered to the reader, e.g., in the SI, if the space is limited.
- Abstract states that XyleE has been used as a model system for GLUTs. As stated also in the introduction later, there are several structures of different mammalian GLUTs already available in different states and bound to different small molecules.
- XyleE is a bacterial protein. I am wondering why it was embedded in a POPC bilayer for the simulations. In proteoliposome experiments, it has been inserted into an E. coli lipid liposome. As indicated in the paper several times, lipids have a large impact on sugar transport rates.
- Some details for the GLUT3 simulations should be provided here, not just simply referring the reader to ref. 19. At the very least it should be stated how many copies of the simulation and how long each have been performed.
- The report of the relevant results on GLUT3 simulations is also missing or at least very terse. Most of the simulations results are on XyleE.
- I am not sure how this was hypothesized before the results: "We hypothesized that perhaps only transported sugars were giving rise to STD NMR signals.". I am not an expert in STD NMR, but without having kinetic data on the rate of transport in these proteins, it seems to me that stable binding of the substrate (or inhibitor) to the binding pocket would equally produce the signal. Although the parallel between the transport assay and NMR signals is pretty convincing of the claim.
- I may disagree with the statement that "both D-xylose and D-glucose sugars were, on average, highly mobile in the sugar-binding pocket (Fig. 3c-d)", based on the figure shown. Xylose looks significantly stable (Fig. 3c).
- The conclusion section reads more like discussion.
- Turn off the hydrogens in the videos, at least for the binding residues, so it is more clear what is happening to the substrate. What are the residues that show up during the simulations? sometimes they are pretty far away from the substrate. I would suggest removing this feature and only show a number of important residues without Hs.

Minor:

- now become silent -> now becomes silent
- Introduction: glucose (GLUT) transporter -> glucose transporter (GLUT)
- Introduction: GLUT transporters -> GLUTs, at multiple places

- Surprisingly, however, despite the wide range of D--glucose binding affinities from 0.007 to 10 mM²: Move the reference 2 to earlier in the sentence since it looks like an exponent to mM.
- found to have a completely enclosed the sugar-binding pocket. -> found to have a completely enclosed sugar-binding pocket.
- very unstable -> unstable
- "v-rescale thermostat and c-rescale barostat" are repeated in Methods.
- XylE and GLUT5 simulations?? 5 or 3?
- lipid metics -> lipid mimetics
- Results: molecular dynamics simulations -> MD simulation
- the longest stable bound state -> the longest stably bound state
- state was observed was for -> state observed was for
- 50% out of the simulation time -> 50% of the simulation time
- we found that in the presence of D-xylose that the -> we found that in the presence of D-xylose the
- Fig 3 caption: center of mass and D-xylose and -> center of mass of D-xylose and
- Fig. 3. the axis label in g/h is cut off, and there is an extra line/box surrounding e. Also, the y axis of rotation seems to be cut off.
- stabilize -> stabilise
- conformationally-stabilise -> conformationally stabilise
- H2-, H3- and H4-protons -> H2, H3 and H4 protons (and similarly in other places)
- C3- and C4-positions -> C3 and C4 positions
- sugar coordinating side-chains -> sugar-coordinating side chains
- spell out a.s.u.
- the absence of a OH-group at the C2-position -> the absence of an OH group at the C2 position
- membrane-bound transporters -> membrane transporters (more concise)
- parasites-to-human -> parasites to human
- well-conserved -> well conserved (adverb not adjective)
- has enabled -> have enabled
- refs. 8 9 10 11 12 13 14-16 17,18 19,20 21,22 -> 8-22
- and sugar inhibitor (D-glucose) in XylE -> and a sugar inhibitor (D-glucose) in XylE
- changes from inward-to-outward states was found -> changes from inward-to-outward states were found
- that sugar substrates catalyses -> that sugar substrates catalyse
- needs only to stabilize -> needs only to stabilise
- catalyzing a substrate -> catalysing a substrate
- transporters catalyzes a conformational change -> transporters catalyses a conformational change

Version 1:

Decision Letter:

Our ref: NSMB-A50400A

5th Dec 2025

Dear Dr. Drew,

Thank you for submitting your revised manuscript "A Two-step Mechanism for Sugar Translocation" (NSMB-A50400A). It has now been seen by the original referees and their comments are below. The reviewers find that the paper has improved in revision, and therefore we'll be happy in principle to publish it in Nature Structural & Molecular Biology, pending minor revisions to satisfy the referees' final requests and to comply with our editorial and formatting guidelines.

Reviewer #2 did not provide comments, but they found the revision adequate and had no further concerns.

We are now performing detailed checks on your paper and will send you a checklist detailing our editorial and formatting requirements in about 2-3 weeks. Please do not upload the final materials and make any revisions until you receive this additional information from us.

To facilitate our work at this stage, it is important that we have a copy of the main text as a word file. If you could please send along a word version of this file as soon as possible, we would greatly appreciate it; please make sure to copy the NSMB account (cc'ed above).

Sincerely,

Katarzyna Ciazynska, PhD
(she/her)
Senior Editor
Nature Structural & Molecular Biology
<https://orcid.org/0000-0002-9899-2428>

Reviewer #1 (Remarks to the Author):

I have read the thorough discussion in the rebuttal letter of the authors, as well as considered the modifications carried out upon consideration of my comments, and I arrived to the conclusion that the manuscript now is in good shape for publication in NSMB. Congratulations for the great work.

Reviewer #3 (Remarks to the Author):

The authors have addressed my comments satisfactorily.

Version 2:

Decision Letter:

3rd Mar 2026

Dear Dr. Drew,

We are now happy to accept your revised paper "A Two-step Mechanism for Sugar Translocation" for publication as an Article in Nature Structural & Molecular Biology.

Your paper will be published online soon after we receive proof corrections and will appear in print in the next available issue. You can find out your date of online publication by contacting the production team shortly after sending your proof corrections.

Authors may need to take specific actions to achieve compliance with funder and institutional open access mandates. If your research is supported by a funder that requires immediate open access (e.g. according to <https://www.springernature.com/gp/open-science/plan-s-compliance> Plan S principles or the <https://www.springernature.com/gp/open-science/us-federal-agency-compliance> NIH public access policy) then you should select the gold OA route, and we will direct you to the compliant route where possible. Because authors warrant under our subscription licensing terms that they haven't committed to licensing any version of their article under a licence inconsistent with the terms of our agreement – including the applicable embargo period – publication under the subscription model isn't suitable for authors whose funders require no embargo.

Sincerely,

Katarzyna Ciazynska, PhD
(she/her)
Senior Editor
Nature Structural & Molecular Biology

<https://orcid.org/0000-0002-9899-2428>

Corresponding authors:

David Drew

We appreciate the positive responses concerning our manuscript. We have carefully examined each comment and have responded to all points below.

Reviewers' Comments:

Reviewer #1 (Remarks to the Author):

The authors report on studies about the question on how GLUT transporters achieve sugar specificity, and propose that using STD NMR spectroscopy it is possible to discern between substrates and inhibitors as only transported sugars produce STD NMR signals. An elegant use of mutant seems to support that STD NMR signals is reporting only on transported substrates. Additionally, the results are discussed in the light of MD simulations and X-ray structure for the binding to PfHT1 transporter and rat GLUT5, supporting that TM7b residues control the formation of occluded state. The work present a high degree of originality in the claim that STD NMR spectroscopy can be a novel tool to characterise transport in sugar porters. The quality of the data and the methodology is high, although the interpretation of the results can be a matter of discussion, as I elaborate below.

In particular, I comment below about all the concerns I find for this work to be published in Nature Structural & Molecular Biology, whereby without a proper response to them I would see difficult to get this work published:

It seems that the claim that STD NMR is a tool to report on sugar transport is something that cannot be expressed as a generalization as it may well be simply a particularity of the study systems. In that sense, I can identify a few drawbacks related to the applicability of the technique for detecting transport that the authors should consider:

- How is it possible that, from ITC, binding is measured for the substrate and the inhibitor to be both in the micromolar range, but the binding of the inhibitor is not detected by STD NMR. That affinity is appropriate for binding detection by STD NMR. Is ITC showing binding but STD NMR not? This is difficult to reconcile for the given affinities.

Thank you for your constructive review. The binding affinities of XyleE to both its substrate (xylose) and inhibitor (glucose) are indeed in a micromolar range according to ITC data

(*Nature* **volume 490**, pages361–366 (2012) *Cell Discovery* **volume 5**, Article number: 14 (2019)). These affinities are collective values of all the interactions between the protein and the respective carbohydrates. Given ITC is an average of different ligand-protein affinities and also measured in detergent, we think the most important data here is the IC_{50} values, which show how well unlabelled sugars compete with the radiolabelled sugar recognition at the sugar binding site. What we find is that the IC_{50} values for xylose vs glucose competition to 3H -xylose uptake is similar and in agreement with the ITC data. Importantly, the double-mutant that is able to transport glucose also has a similar IC_{50} for glucose as the WT protein. Other methods, such as SSM-based electrophysiology, have also calculated similar affinities for xylose and glucose interactions to Xyle in liposomes for the WT protein (glucose EC_{50} = 0.4 mM; xylose K_M = 1.7 mM) as well as for the double Q175I/L297F mutant (glucose K_M = 1.5 mM; xylose K_M = 1.4 mM) (<https://www.jbc.org/action/showPdf?pii=S0021-9258%2821%2901315-6>).

K_M for xylose is between 0.1 to 0.5 mM

K_d for xylose by ITC: 0.35 mM

K_d for glucose by ITC: 0.77 mM

K_M for xylose by SSM: 1.7 mM

EC_{50} for glucose by SSM: 0.4 mM

Therefore, it is not a question of differences in affinities between these two sugars. Why doesn't glucose give rise to STD NMR signals? By locking Xyle in either the outward- or inward- facing conformation, we can show that the signal for xylose binding now becomes silent. Clearly, xylose is a transported sugar and not an inhibiting sugar, like glucose. ITC and structural data of these mutants with xylose show that these mutants have retained strong binding for xylose (*Cell Discovery* **volume 5**, Article number: 14 (2019)). We cannot assess these mutants by competitive transport, but with and without the addition of DTT to the **same liposomes** we observe a clear correlation between the STD NMR signal and transport. We have now included data showing that the STD NMR signals for xylose also become silent upon the addition of glucose.

To support these arguments I think it is important to also reflect on the differences in STD NMR signals for Xyle protein incorporated into liposomes made from either *E. coli* vs brain lipids. In this setup there is **no difference** made to the sugar binding site and the reconstitution efficiency is the same. Therefore, if it was a simple binding reaction it should show similar STD NMR signals. However, again, we see a clear correlation between the STD signal and the amount of transported sugar. Taken together, we have concluded that only transported sugars give rise to STD NMR signals.

The only difference between a transported sugar and one stabilised by either the outward-occluded or inward-occluded state is the later can pass through the **occluded** state. We conclude that the occluded state must have the strongest coordination for the sugar. To support this conclusion, we have performed MD simulations. What we observe is that xylose and glucose are both fairly mobile in the outward-occluded state. In one out of the five simulations for xylose, however, the sugar is coordinated for the entire length of the simulation, which corresponds to closure of the extracellular gate TM7b. In comparison, glucose is less likely to be coordinated and, further, TM7b always remains shut, i.e., consistent with its annotation as a sugar inhibitor. Unfortunately, the time scales in these MD simulations are too short to sample sugar translocation through the occluded state. Previously, we used enhanced MD simulations to reconstruct the transport cycle of GLUT5. These enhanced simulations took considerable effort (3+ years) and so it's not an approach we can easily add to this study here (eLife (2023);12:e84808). Nonetheless, what we observed in these simulations was that fructose only became highly coordinated when it transitioned into the occluded state.

Indeed, the coordination of fructose in GLUT5 matches how 2,5-dihydromannitol is bound in PfHT1, which has a similar K_M for fructose as GLUT5.

McComas S et al. eLife (2022)

We thus propose that non-transported sugars do not give rise to STD NMR signals as they are never bound for a long enough period of time to receive sufficient saturation transfer -estimates are at least 100 μ s is required (J Am Chem Soc 123, 6108-6117 (2001). Effectively, all other states, apart from the occluded state, become silent. Sugar porters have fairly weak binding affinities (low mM) range, which is already at the detection limit for STD NMR. We think the fact STD NMR only detects transported sugars could be specific to transporter's, like GLUTs, where the substrate affinities are weak.

Lastly, In support of the evidence that previously determined partially occluded states in sugar porters represent suboptimal coordination, we have now included the cryo EM structure of PfHT1 in complex with 2,5 AHM. What we observe by cryo EM is that 2,5-AHM is not fully coordinated in the inward-occluded state as it does not interact with the residue W412. We could previously show that the W412A mutant did not transport this sugar and had weak STD NMR signals. The position of 2,5-AHM also matches the position of urate in the recent inward-occluded structure of GLUT9 determined by cryo EM as well as the position of the glucoside sugar from the detergent β -NG in the first crystal of GLUT1. In both cases the sugar is not fully coordinated as it lacks the critical interaction with the residue corresponding to W412 in PfHT1.

By now comparing the inward-occluded and occluded states for the **same** transporter for the first time we can observe that the sugar adjusts its position by 2\AA and becomes more highly coordinated in transition to the occluded conformation. In light of this new structural information we re-analysed all determined structures of substrate bound sugar porters. What we find, in fact, is that in all crystal structures the partially occluded states have also been co-crystallized with either a detergent or lipid-like molecule. In other words, we think these co-ligands and/or the low temperatures have been able to stabilize the substrate in these suboptimal substrate-bound states. The exception is GLUT9, which like PfHT1 was determined by cryo EM and also in the inward-occluded conformation.

Its also worth mentioning that the inward-occluded state observed by cryo EM is also the most favourable conformation in the free energy landscape we constructed for GLUT5 in the presence of fructose.

McComas S et al. eLife (2022)

Taken together, whilst I think we bring the use of STD NMR to the transporter community, I think the main message of this study is that we need to be careful with interpreting substrate-bound structures of transporters, which have all been obtained at low temperatures. In other words, we “really” need to understand protein dynamics to ascertain how substrates are both selected for and translocated. Here for the first time we can demonstrate that similar to soluble enzymes, the transition state has the highest affinity for the substrate.

- If the application of STD NMR to distinguish transport or inhibition is simply based on the interpretation of a "significant" reduction of the STD responses in the case of the inhibitor, the methodology seems extremely hard to be properly applied or, better to say, the results properly interpreted. In that sense: what is a "significant" reduction? STD NMR is affected by different things, related not only to the kinetics of the thermodynamics of the process of binding, but also related to differences in sample preparation (e.g. it would be nice to repeat the study of xylose binding with different proteoliposome prepared samples, to check the reproducibility of the STD level of intensities).

The ligand to protein concentration is kept constant and the STD NMR signal is consistent between different preparations.

For transported sugars, the STD NMR signals of mutants (PfHT1, XyleE, GLUT5) mostly match the coordination of sugars seen in the crystal structures, i.e., apart from W412 in the new cryo EM structure of PfHT1. Indeed, STD NMR has shown that only the furanose form of the sugar binds to PfHT1, which we could confirm with transport assays of 2,5-dihydromannitol.

We think the XyleE mutants that the double Q175I/L297F mutant that can transport glucose is an important control, as it now produces STD NMR signals. Further, we can show that in the XyleE CC mutant that both STD NMR and transport uptake becomes

dependent on reduction by DTT. These experiments are performed with the same proteoliposomes.

We agree there is an interpretation of when to decide an STD-NMR signal is produced, and differing from “noise”. We would argue that the included spectrograms enables the reader to review the relative differences between a transported and inhibiting carbohydrate.

The concentration of Xyle in proteoliposome was carefully quantified by fluorescence using a Xyle-GFP fusion.

- Unless the values of the STD NMR intensities are given, and a “factor” quantifying the differences in level of intensities between substrates and inhibitors, it looks as if the interpretation is in the eye of the beholder (...is it an observed reduction in STD NMR significant as to claim that there is no transport?). Again, it’s worth mention that comparing different samples is tricky as during the preparation process the final sample conditions might differ leading to different STD responses. Could competition experiments on a single proteoliposome sample determine if one ligand is observed and not the other? For example, a sample of Xyle containing xylose could be put in competition with increasing amounts of glucose, and viceversa.

Thanks for these suggestions. In this work, we used integration of NMR spectral region corresponding to sugars to acquire the intensity of each spectrum.

Xylose	1H intensity	STD intensity
Wild-type	7368907392	2689704992
V35C/E302C reduced	8589200704	2101279136
V35C/E302C oxidised	8484326912	917635040
G58W/L315W	9470401280	54247744
Q175IL297F	10426632320	1962936512
Wild-type in brain fraction 7 lipid	6347163437	803366628

Glucose	1H intensity	STD intensity
Wild-type	16814787840	454581456
Q175IL297F	12288783936	812448200
Wild-type in brain fraction 7 lipid	7382777782	184619244

Supplementary table 4. 1H and STD intensities.

As previously noted, we now include STD NMR spectrum of glucose added to proteoliposomes that had given a strong signal for xylose.

- I cannot see why binding to these receptors, even in the absence of transport, can never be appropriate for STD NMR observation (in fact that is what I would expect for the studied system, taking consideration of the affinities measured by ITC).

This is essentially the same question as was asked previously. We agree that the lack of STD NMR signals for glucose is a surprise in XyleE, but we think our controls support our conclusions. Remember, that for these transporters the binding site for glucose is conserved from parasites to human yet the kinetics and substrate specificity is very different.

Human $K_M = 1.3$ mM
 GLUT3 *Specialist for glucose transport into neurons*

Parasite $K_M = 0.9$ mM
 PfHT1 *Malarial parasite transporter for many different sugars*

Plant $K_M = 0.005$ mM
 STP10 *Plant transporter for glucose*

E. coli D-Glucose is an inhibitor
 XyleE *Bacterial transporter for xylose*

- The concentrations of the ligands are low for STD NMR standards. To be on the safe side about the claim that inhibitors that do not produce transport do not give STD NMR signals, I would recommend running experiments with at least 1 or 2 mM of substrate or inhibitor. In low concentrated ligand samples, STDs might be obscured by the spectral noise, something that would be much improved under larger ligand concentrations. That is the main reason why I propose that the authors should provide a table with the measured STD intensities.

Thank you for the advice. We did included the data recorded at a substrate concentrations of 0.5mM. We would expect inhibitors to produce STD NMR signals and have rather clarified we were probing the differences between **sugar** inhibitors and **sugar** substrates rather than small molecule inhibition, which we think is a different question all together.

The tables are now provided with the intensities. We have updated the manuscript accordingly.

Xylose	1H intensity	STD intensity
Wild-type	7368907392	2689704992
V35C/E302C reduced	8589200704	2101279136
V35C/E302C oxidised	8484326912	917635040

G58W/L315W	9470401280	54247744
Q175IL297F	10426632320	1962936512
Wild-type in brain fraction 7 lipid	6347163437	803366628

Glucose	¹ H intensity	STD intensity
Wild-type	16814787840	454581456
Q175IL297F	12288783936	812448200
Wild-type in brain fraction 7 lipid	7382777782	184619244

- Figure 1f: what do the authors mean here by “normalized STD effect”? How can it be 0-100 scale for xylose in liposomes from E.coli and only close to 40% in brain lipids fraction? I do not understand how the scales can be transferable between so different sample conditions. Even if the authors want to show that STD responses are much higher in liposomes than in brain lipids, that cannot be correlated directly to transport, as both samples are so different in nature that one would expect, in fact, to get very different STD NMR responses (e.g. the saturation of the proteins can be different using the same irradiation frequency, due to differences in particle sizes, compacity, etc...). This concern is also transferable to the right column bar graphs in Figure 2.

We agree the absolute values are needed for a direct comparison, and we have now included the respective values in Supplementary Table 1. Presenting the normalized STD NMR effect values in the figures next to the transport activity enables an easier comparison to the transport activity.

The composition of sample (mainly lipids in this case) can introduce differences in NMR results. Therefore, we compared the correlation between the NMR results and transport activity. We think this is an important point as it supports the conclusion that only transported sugars give rise to STD NMR signals.

- It is necessary to run all the STD NMR experiments on "blank samples" as controls (samples of liposomes devoid of proteins), under increasing concentrations of ligands (in particular after the proposal from this reviewer of increase the ligand concentrations in the experiments). This is to discard the possibility of having different proportions of non-specific binding to the liposomes by Glc or Xyl that might be impacting the STD results.

Yes, this had been done. We have added a figure with spectrum of protein free samples (Fig. S4).

Supplementary Figure 1b. The 1D ¹H spectrum of protein free liposome samples per sugar and lipid combination.

- Comparing the STD NMR spectra of glucose binding in figures 1e and 2c, one might get the wrong conclusion that the interpretation of binding in the figure 2c and not binding in figure 1e is just a matter of zooming the Y-axis of the spectra. This is again the reason why it is fundamental that the authors provide tables with the experimental intensities of the STD NMR spectra, to evaluate the ability for discrimination of the proposed approach between substrates and inhibitors.

We now have included a table (Supplementary Table 1) of these intensities for a direct comparison.

- Figures 2a and 2b: why do not the authors provide also the studies with Glc, to keep consistency with all the work presented?

We actually dont see the relevance for this experiment, since these mutations do not transport glucose and we have already shown that WT does not respond to glucose.

- MD simulations were carried out on outward-occluded states of the receptor. I propose also running the MD simulations with the outward-open state of the receptor, as it may well be responsible also for STD NMR responses and corresponding differences in the observed intensities (e.g. Glc being more unstable in that state?).

Outward-open state will probably not have sufficient interactions with either xylose or glucose to be detected by STD-NMR, as the respective ligands are free to leave the pocket and outlive the remaining simulation time tumbling freely in the surrounding solvent.

We thus opted to use the outward occluded state due to the fact that in the outward open state the sugar is not fully coordinated and leaves quickly the pocket. This would result in simulating an empty transporter after the initial release. Moreover, we noticed how the outward open conformation in absence of sugar reverts to the outward occluded state, so we performed simulations of the double tryptophan mutant (G38W/L315W) described in <https://doi.org/10.1038/s41421-019-0082-1>, annotated to be an outward open state, but still capable of binding xylose. The resulting gate dynamics and probabilities are reported below. As you can see, even the double mutant revert to the outward occluded state in absence of sugar.

- How do the authors confirm that all the receptors in the proteoliposomes are well oriented for the transport?

Please note that these are “transporters” and not “receptors”. From a functional point of view these are very different and this distinction does reflect how one interprets the data, i.e., receptors are stabilised in the G^* active state by typically a tight ligand-protein interaction, whereas transporters are like enzymes, but need to move their substrate across the membrane, which means they need weaker (substrate) affinities as they also need to allow the substrate to be released in a different conformation again.

Transporters can work in either direction, which is based on the electrochemical potential of the substrate. Further, since only transported sugars give rise to STD NMR signals it should not matter the exact population of inside-out vs right-side out proteins in liposomes. We have, however, measured this previously for GLUT5 and PfHT1 and found that their orientation is 90% and 20% right-side out respectively (Nature Comms 14: 4070). For PfHT1 with a low right-side orientation, its kinetics (K_m) however is the same as that measured from in cells, which makes sense as the transporter needs to sample all conformations in the transport cycle even if its starts from a different location.

Lines 202 – 205: “Rather than by difference in substrate affinities, we conclude that low intensity STD NMR signals are a result of glucose failing to stabilize an occluded conformation in XyleE, which would allow for a long enough interaction time with coordinating residues for sugar protons to achieve sufficient saturation transfer.” And also Lines 322 – 23: “Rather, from STD NMR analysis and MD simulations we can conclude that subtle differences enable a substrate sugar (D-xylose) to be coordinated for a longer period of time than glucose, by which D-xylose can better engage with TM7b”.

This reasoning poses a fundamental thermodynamics/kinetics question: as I understand, in fact, a longer interaction time is associated to a higher affinity. If, for a given ligand to a receptor the STDs are stronger because of a longer "contact time", the affinity for that ligand should accordingly be stronger, however, both, substrate and inhibitor show rather similar affinities. There is something fundamental related to the claim that I cannot understand.

The longer interaction is associated with a higher affinity would be correct if the ligand was sampling only one conformation. However, the ligand (substrate) needs to induce the transition to the occluded state. At least between glucose and xylose in XyleE, the ability for a substrate to catalyse this transition is not because it has a higher affinity, but

because it can engage correctly with the extracellular gate. The STD NMR signals for xylose become silent when the protein can no longer access the occluded state.

In support of our measurements are some earlier ^1H NMR glucose interactions made to GLUT1 in red blood cells (PNAS. 83:3277-3281). That binding and dissociation of glucose from the carrier are much faster than the overall maximum rates of transport.

Proc. Nati. Acad. Sci. USA. Vol. 83, pp. 3277-3281

They concluded that there are many "fast" sugar-protein interactions, but only 10% of these interactions leads to a successful translocation event. In light of our own data, the fast sugar binding events would be the sugar binding to the outward-occluded state and the longer time scales are the events through the occluded state. In other words, not all sugar-binding events are successful to catalyse transport, which would be consistent with our MD simulations in Xyle.

Lastly, thermodynamic measurements have shown that GLUT1 and Xyle are entropy-driven reactions (Biochimica et Biophysica Acta 901:229-238 (1987)). Rather than an enthalpic interaction energy for a tight ligand-protein interaction, there isn't much energy gained from sugar binding, but rather the sugar likely displaces water from the solvent exposed cavity and the removal of water in forming the sugar-protein complex.

THERMODYNAMIC PARAMETERS FOR THE TRANSPORT SYSTEMS FOR GLUCOSE AND L LEUCINE IN RED BLOOD CELLS

Standard enthalpies and entropies were calculated as described in the text. Basic Gibbs free energies and transition states present were calculated for a glucose concentration of 5 mM or a leucine concentration of 0.2 mM. C_o and C_i represent outward-facing carrier states, while G and L represent glucose and leucine.

Glucose carrier	ΔH° (kJ·mol ⁻¹)	ΔS° (J·K ⁻¹ ·mol ⁻¹)	Basic Gibbs free energy change (kJ·mol ⁻¹)		Carrier state
			0 °C	37 °C	
Substrate binding to the					
outward-facing carrier	5.51 ± 3.39	58.1 ± 24.0	1.7	1.2	C _o
inward-facing carrier	-4.95 ± 3.73	17.3 ± 21.9	2.3	3.3	C _i
Carrier reorientation from inside to outside					
unloaded carrier	46.0 ± 5.69	145 ± 21.0	6.4	1.05	C _o G
carrier-glucose complex	56.3 ± 8.01	185 ± 30.7	5.8	-1.05	C _i G

Biochimica et Biophysica Acta 901:229-238 (1987)

Consistent with thermodynamic parameters calculated from kinetics by the Arrhenius equation, thermodynamic calculations from ITC data shows that xylose binding to XylE is largely entropy driven $\Delta S = 25$ kcal.mol. However, the XylE WW mutant trapped between the outward to outward-occluded state is associated with a much smaller $\Delta S = 7$ kcal.mol. We think this data supports the argument that the occluded (transition-state) formation — accessible during sugar translocation — is largely entropy driven and is likely because water are removed from the sugar binding site. Indeed, locked mutants have a negative ΔH , which suggest that these semi-occluded states have already released water as compared to the apo state. This is something we would like to explore in future experiments.

Tabel S1 | Thermodynamic parameters of ITC test

XylE	K_a (M ⁻¹)	ΔH (kcal·mol ⁻¹)	ΔG^1 (kcal·mol ⁻¹)	ΔS (cal·mol ⁻¹ ·k ⁻¹)	$T\Delta S^2$ (kcal·mol ⁻¹)
Wild type	1.97x10 ³	2.95	-4.45	25.1	7.40
Crosslinked EndoCC	3.96x10 ⁴	-1.38	-6.21	16.4	4.83
Crosslinked ExoCC	1.59x10 ³	-0.81	-4.32	11.9	3.51
XylE-WW	2.65x10 ⁴	-3.87	-5.97	7.12	2.10

1. $\Delta G = -RT \ln K_a$.

2. $T\Delta S = \Delta H - \Delta G$.

Cell Discov 5, 14 (2019)

Line. 209:

Why do the authors use the word “coupling” between glucose and fructose?

This is a transporter terminology in reference to the fact that the substrate binding is “coupled” to local, gating rearrangements that in this case is TM7b.

Line 215:

Why “Suppl. Fig. 2a” is mentioned here?

The reference to supplementary figure 2a was included to draw attention to the SEC-analysis of the sample quality.

Line 217: “more robustly” is not a proper wording for describing the presence of strong STD signals. I would recommend “more closely”

We updated the sentence omitting the word “robustly”.

Lines 218 – 219: “The STD NMR analysis is further consistent with the hydrogen bonding distance between the protons in D-glucose and the sugar coordinating side-chains.”

This is interesting, as it seems that they are referring to the orientation in the occluded state, isn't it? If that is the case, how can STD NMR results be reporting on transport, but the epitope reported is related to the binding site in the occluded state? In line with this concern, I would recommend the authors to try to measure binding epitopes for inhibitor, even if the intensities are very low, as it might well be that the resulting epitope will equally be reporting on the binding as observed in the 3D structures. Reword to clarify and justify why not to do epitope mapping.

There seems to be a misunderstanding. We have clarified why we think the signals are poor for the “sugar” inhibitor glucose, but “small molecule” inhibitors should give strong STD NMR signals if they bind tight enough to the transporter. We have recently used STD NMR to probe binding to an ATP/ADP exchanger (Nature (2020) 578 (7794):321-325).

Lines 264 -266: “Alanine substitutions of the sugar coordinating residues Q305, Q306, N311, N341 all showed low intensity STD NMR signals Fig. 5a consistent with transport assays that have demonstrated these variants abolished D-glucose transport”

Again, this is a clear case were the claim that low intensity indicates no transport doesn't seem to be so strong taking into consideration the evidences in Figure 5a (again, what is

the meaning of “normalized STD effects”? (how is scale transferable between samples?). What is more, in Suppl. Fig. 3, the alanine substituted mutants seems to show clear STD NMR responses. Given these evidences, the claim that STD NMR reports on transport doesn't seem to be clearly supported.

We disagree. These are low STD NMR signals compared to the WT protein as can be visually seen by the STD NMR traces. Transport activity is very low, but not completely dead either. If we measured transport for hours (rather than 1 min), as in the STD NMR experiments, then we might see some low accumulation of sugar. We don't think these additional experiments are needed in this case.

Reviewer #2 (Remarks to the Author):

The authors focused on studying sugar transporters for many years and have published several important papers on the glucose transport mechanism. This work explores the conformational selection-fit model for sugar transport to elucidate how glucose transporters (GLUTs) achieve sugar specificity. The authors employ saturation transfer difference (STD) NMR spectroscopy to analyze the differences between sugar substrates and inhibitors, revealing the molecular basis for the transport mechanism. The study highlights the importance of specific residues in controlling the formation of the occluded state, concluding that the optimization of substrate interactions with this state primarily determines substrate specificity in transporters. Before accepting, the authors should address the following concern

1. The author hypothesized that "only transported sugars were giving rise to STD NMR signals." Some sugars that cannot be bound/transported by XylE should be considered for measurement by STD NMR as additional controls.

Glucose is a well-known sugar inhibitor of XylE. The question was to determine the difference between a substrate sugar and a substrate inhibitor. This is a fundamental question in transport biology that I think we have been able to answer. I don't know of other sugar inhibitors for XylE and if the sugars cannot be bound/transported than they will not give STD NMR signals and so it wasn't clear to us what the outcome of these experiments would prove.

2. The transport assays in Fig2a and Fig2b contain two almost identical WT groups (control). Please recheck it. The authors should provide data from the two individual experiments in the two graphs.

Yes, this is the same data as the “WT” reference has been measured from multiple reconstitutions. We have now updated the WT for Figure 2b, clearly stating its taken from Figure 2a.

3. The author makes a detailed explanation in the conclusion. However, the difference between conformational selective-fit model and previously accepted models is less clear.

I am not sure if there is an “accepted” model, but indeed we had previously proposed a conformational selective fit model based on structures of sugar porters in different conformational states (Nature 578:321) Nonetheless, there is a large difference between proposing a model based on structures and proving this model, which requires experimental and computational dynamics that is able to link the different conformational states (at least those that are long enough lived that they could be captured by experimental structures). In fact, we now update this model by including cryo EM structure of PfHT1 that shows these pre-bound inward-occluded states lack full coordination of the substrate. We think that previous inward-occluded and outward-occluded states have “looked” like stably coordinated sugars, but had been partially stabilised by either detergent or lipid-like molecules binding to either TM7 and TM10 gating helices. In light of this new cryo EM experimental data, MD simulations, NMR data and consideration of thermodynamics we update our mechanism as a two-state model. First, conformational selection of gate closure to a sugar bound (non-optimal) state followed by a induced-fit driving the rocker-switch transition whereby the transporter accesses a occluded, transition state, that has the highest affinity for the substrate sugar.

4. The resolution of the crystal structure is slightly low. The local density of small molecules is also not very good. I'm worried that the small molecules are facing the wrong orientation. Can authors collect more diffraction data sets and solve the structure with higher resolution?

With 4 molecules in the a.s.u and solvent flattening, we think the map quality of the bound sugar is very good. Since 2,5-DHM is a symmetrical molecule this is the only reasonable pose. Nevertheless, we repeated the structural studies by cryo EM with maps reconstructed to 2.4 Å. The cryo EM supports the sugar pose modelled in the crystal structure, **but** also we obtain a different conformation in which the sugar has not been fully coordinated by interaction with the tryptophan residue W412.

Reviewer #3 (Remarks to the Author):

The manuscript by Ahn, et al. describes primarily STD NMR results to conclude that sugar specificity in glucose transporters (GLUTs) is mainly controlled by the ability of the substrate to form an occluded state of the transporter, not by their binding affinity to the protein. They use several proteins for their measurements, including Xyle, a bacterial homolog of GLUTs, which is actually the main system studied here. MD simulations are also used to make sense of the data. MD results show that glucose in the binding site of Xyle locks the protein whereas xylose produces the occluded state, though this observation is not consistent, and in some cases the sugar substrate leaves the binding site. The study is complemented by structural determination of another bacterial homolog. In general, the study reports interesting results, and I lean towards recommending it for publication, but the presentation is rather poor and requires major

revision. Below are some specific comments about the manuscript.

- As I indicated, the text is a mess and needs critical reading and revision for typos and grammatical errors. Some of these have been mentioned below in my minor comments.

- At points not enough information is provided to get a clear idea about what has been done, particularly in the methods section. More details can be offered to the reader, e.g., in the SI, if the space is limited.

We thank the reviewer for the suggestion to make our simulations more understandable and reproducible by having a more exhaustive methods section, we have worked on adding more details and we hope that this is now clearer

- Abstract states that XylE has been used as a model system for GLUTs. As stated also in the introduction later, there are several structures of different mammalian GLUTs already available in different states and bound to different small molecules.

We used XylE as a model system first and foremost in our NMR experiments as we have a clear example for a sugar porter that recognises both a sugar inhibitor and a sugar substrate. There are probably GLUT examples too, but none that are definitive. So we focused our MD simulations on the same system in order to provide clarity in the interpretation of our NMR analysis.

- XylE is a bacterial protein. I am wondering why it was embedded in a POPC bilayer for the simulations. In proteoliposome experiments, it has been inserted into an E. coli lipid liposome. As indicated in the paper several times, lipids have a large impact on sugar transport rates.

We agree with the reviewer that lipids have a large impact on the sugar transport rates, but in these classical MD we do not observe the entire transport due to the timescale limitations of the method. Since the initial objective was to compare XylE to the GLUTs we previously studied (GLUT5, GLUT3 and PfHT1), we opted to keep the same lipid in all simulations, since we did not intend to study transport in these simulations and the simulation time would not allow for that. Even in experiments, POPC is good enough for binding measurements and perhaps for transport too, albeit at slower rates (e.g eLife 12:e84808. Moreover, we wanted to compare XylE and GLUT3, so we did not want to add more variables by adding different lipids and charged lipids at that.

The charged lipids would also complicate the intrinsic inability of classical MD. We agree, however, that to study the full transport cycle more complicated lipid compositions should be used and we are working to do so in a paper dedicated to a full transport cycle of XyleE.

- Some details for the GLUT3 simulations should be provided here, not just simply referring the reader to ref. 19. At the very least it should be stated how many copies of the simulation and how long each have been performed.

We have now added all the fundamental details for the simulations of GLUT3, while still referring to the original publication for further details. Moreover, we have repeated the GLUT3 simulations using the same parameters we used for XyleE and we now report this in the supplementary figure as well (Supplementary Figure 3).

- The report of the relevant results on GLUT3 simulations is also missing or at least very terse. Most of the simulations results are on XyleE.

We agreed with the reviewer that the description of the relevant results in GLUT3 was quite terse, we have now added a more detailed description in the text.

- I am not sure how this was hypothesized before the results: "We hypothesized that perhaps only transported sugars were giving rise to STD NMR signals.". I am not an expert in STD NMR, but without having kinetic data on the rate of transport in these proteins, it seems to me that stable binding of the substrate (or inhibitor) to the binding pocket would equally produce the signal. Although the parallel between the transport assay and NMR signals is pretty convincing of the claim.

- I may disagree with the statement that "both D-xylose and D-glucose sugars were, on average, highly mobile in the sugar-binding pocket (Fig. 3c-d)", based on the figure shown. Xylose looks significantly stable (Fig. 3c).

Indeed we agree that the figures shown did not reflect the statement. There was a mistake in the picture we showed in the initial submission, we have now corrected the mistake. Moreover, we added two panels in Figure 3 that show better the coupling between sugar binding and gate closure, in which we report the probability density for the gate distance for structures presenting the sugar in the crystal-like (or starting) position or in other poses. Moreover, we added a Supplementary Fig. 2, that shows

the gate time course and the corresponding cluster structures for each of the individual simulations.

- The conclusion section reads more like discussion.

- Turn off the hydrogens in the videos, at least for the binding residues, so it is more clear what is happening to the substrate. What are the residues that show up during the simulations? sometimes they are pretty far away from the substrate. I would suggest removing this feature and only show a number of important residues without Hs.

We thank the reviewer for the suggestion, indeed we agree that the videos were not very clear. We opted to change perspective and keep only the coordinating residues always on (now without hydrogens). We hope these videos are easier to interpret.

Minor:

- now become silent -> now becomes silent

Text has been adjusted.

- Introduction: glucose (GLUT) transporter -> glucose transporter (GLUT)

Updated

- Introduction: GLUT transporters -> GLUTs, at multiple places

Updated to reduce repetition

- Surprisingly, however, despite the wide range of D--glucose binding affinities from 0.007 to 10 mM²: Move the reference 2 to earlier in the sentence since it looks like an exponent to mM.

The Km values have been placed within a parenthesis, followed by their reference

- found to have a completely enclosed the sugar-binding pocket. -> found to have a completely enclosed sugar-binding pocket.

This sentence was reformulated to clarify.

- very unstable -> unstable

The word "very" was removed.

- "v-rescale thermostat and c-rescale barostat" are repeated in Methods.

The repetition was removed.

- XylE and GLUT5 simulations?? 5 or 3?

The text was updated to clarify the simulation numbers.

- lipid metics -> lipid mimetics

Text was corrected.

- Results: molecular dynamics simulations -> MD simulation

Abbreviation used instead.

- the longest stable bound state -> the longest stably bound state

The text has been updated accordingly.

- state was observed was for -> state observed was for

Surplus "was" was removed.

- 50% out of the simulation time -> 50% of the simulation time

The word "out" was omitted.

- we found that in the presence of D-xylose that the -> we found that in the presence of D-xylose the

Surplus "that" removed.

- Fig 3 caption: center of mass and D-xylose and -> center of mass of D-xylose and

Text changed accordingly.

- Fig. 3. the axis label in g/h is cut off, and there is an extra line/box surrounding e. Also, the y axis of rotation seems to be cut off.

The figure 3g/h has been replaced.

- stabilize -> stabilise

Stabilize does adhere to the American English style of Nature, hence this spelling was kept.

- conformationally-stabilise -> conformationally stabilise

Surplus hyphen was removed.

- H2-, H3- and H4-protons -> H2, H3 and H4 protons (and similarly in other places)

The surplus hyphens have been removed.

- C3- and C4-positions -> C3 and C4 positions

The surplus hyphens have been removed.

- sugar coordinating side-chains -> sugar-coordinating side chains

The hyphens have been changed.

- spell out a.s.u.

Abbreviation spelled out.

- the absence of a OH-group at the C2-position -> the absence of an OH group at the C2 position

The surplus hyphen was removed.

- membrane-bound transporters -> membrane transporters (more concise)

Text updated accordingly.

- parasites-to-human -> parasites to human

The surplus hyphens were removed.

- well-conserved -> well conserved (adverb not adjective)

The surplus hyphen was removed.

- has enabled -> have enabled

Text was updated accordingly.

- refs. 8 9 10 11 12 13 14-16 17,18 19,20 21,22 -> 8-22

Updated

- and sugar inhibitor (D-glucose) in Xyle -> and a sugar inhibitor (D-glucose) in Xyle

The text has been updated to specifically state " the sugar inhibitor (D-glucose) in Xyle"

- changes from inward-to-outward states was found -> changes from inward-to-outward states were found

Text updated accordingly.

- that sugar substrates catalyses -> that sugar substrates catalyse

Text updated accordingly.

- needs only to stabilize -> needs only to stabilise

Stabilize does adhere to the American English style of Nature, hence this spelling was kept.

- catalyzing a substrate -> catalysing a substrate

Catalyzing does adhere to the American English style of Nature, hence this spelling was kept.

- transporters catalyzes a conformational change -> transporters catalyses a conformational change

Catalyzes does adhere to the American English style of Nature, hence this spelling was kept.